# TeNet: Text-to-Network for Compact Policy Synthesis

## Abstract

Large vision-language-action (VLA) models such as PaLM-E, SayCan, and RT-2 enable robots to follow natural language instructions, but their billions of parameters make them impractical for high-frequency real-time control. At the other extreme, compact sequence models such as Decision Transformers are efficient but not language-enabled, relying on trajectory prompts and failing to generalize across diverse tasks. We propose TeNet (Text-to-Network), a framework that bridges this gap by instantiating lightweight, task-specific policies directly from natural language descriptions. TeNet conditions a hypernetwork on LLM-derived text embeddings to generate executable policies that run on resource-constrained robots. To enhance generalization, we introduce grounding strategies that align language with behavior, ensuring that instructions capture both linguistic content and action semantics. Experiments on state-based Mujoco and Meta-World benchmarks show that TeNet achieves robust performance in multi-task and meta-learning settings while producing policies that are orders of magnitude smaller. These results position language-enabled hypernetworks as a promising paradigm for compact, language-conditioned control in state-based simulation, complementary to large-scale VLAs that tackle vision-based robotics at massive scale.

## 1 Introduction

Recent breakthroughs in large language models (LLMs) such as GPT (Brown et al., 2020) and LLaMA (Touvron et al., 2023) have demonstrated remarkable generalization across diverse tasks and domains. In robotics, vision-language-action (VLA) models such as PaLM-E (Driess et al., 2023), SayCan (Brohan et al., 2023), RT-2 (Zitkovich et al., 2023), OpenVLA (Kim et al., 2025), and OCTO (Team et al., 2024) extend this paradigm, conditioning robot behavior on natural language and visual inputs. These systems point toward an exciting future where robots can flexibly follow human instructions.

Yet, their practicality is limited. State-of-the-art VLAs often contain billions of parameters, making inference too slow for high-frequency control loops and exceeding the hardware constraints of mobile robots. On the other end of the spectrum, compact models such as Decision Transformers (DT) (Chen et al., 2021) and Prompt-DT (Xu et al., 2022) are lightweight and efficient, but not inherently language-enabled. They rely on trajectory prompts rather than natural instructions, require demonstrations even for unseen tasks, and degrade sharply as the number of tasks increases. This trade-off leaves a gap between large but impractical VLAs and compact but non-language-grounded sequence models. In this work, we deliberately restrict ourselves to low-dimensional, state-based benchmarks so as to isolate the contribution of language-conditioned policy instantiation, and we do not address perception or vision.

Several works have attempted to bridge this gap by using LLMs in indirect ways. Code-as-Policies (Liang et al., 2023) translates instructions into robot API calls, while Code-as-Rewards (Venuto et al., 2024) leverages vision-language models to automatically translate task descriptions into reward signals for reinforcement learning. These approaches creatively connect language and control, but they depend on predefined interfaces or exact simulators, making them difficult to deploy in real-world robotics.

In contrast, we ask whether language itself can serve as the direct conditioning signal for policy instantiation. Rather than running a large model inside the control loop, we use it once – at policy

instantiation –through a hypernetwork (Ha et al., 2016). Our framework, **TeNet (Text-to-Network)**, conditions a hypernetwork on LLM-derived text embeddings to generate compact task-specific policies that can run onboard resource-constrained robots. This enables direct text-conditioned policy instantiation at inference time, without requiring trajectory prompts or demonstration replay.

While direct text-to-policy generation demonstrates that compact policies can indeed be synthesized from language alone, we find that effectiveness improves significantly when language is *grounded in behavior*. To achieve this, we align text and trajectory embeddings using two strategies: direct embedding alignment (MSE) and contrastive objectives. This grounding ensures that task descriptions capture not only linguistic content but also behavioral semantics, enriching language representations with trajectory structure. As a result, TeNet achieves stronger generalization across tasks and improved performance in multi-task and meta-learning settings.

Our goal is not to surpass large VLAs – which target vision-based benchmarks at massive scale – but to open a complementary direction: *language-enabled hypernetworks for compact policy synthesis in state-based simulation*. We restrict ourselves to trajectory-based domains (Mujoco and Meta-World) as a necessary first step, systematically testing whether compact policies generated from text and grounded in trajectories can provide robust multi-task performance. TeNet is therefore complementary to VLAs rather than a competitor: it focuses on efficient policy instantiation in state-based domains, not on solving end-to-end vision-language control. TeNet introduces the first framework that uses natural language only once—as a conditioning signal for a hypernetwork that *generates* a compact, fully executable policy. After instantiation, the resulting controller operates independently of any language model, receiving only states and running at high frequency.

In summary, our contributions are:

- **Text-to-Network Policy Generation.** We introduce TeNet, a framework that conditions a hypernetwork on LLM text embeddings to synthesize compact, task-specific robot policies. Language is used only once—as a conditioning signal for the hypernetwork to generate all policy parameters. The resulting controller is a standalone ∼40K-parameter network that receives only states at inference and runs at high frequency without any language model or multimodal processing.

- **Grounding Language in Behavior.** We adopt standard alignment strategies to map text and trajectory embeddings – including direct embedding alignment and contrastive objectives – which enrich language representations with behavioral semantics and improve generalization in multi-task and meta-learning. These grounding mechanisms are standard tools and serve as auxiliary components: they enhance robustness but are not the core novelty of TeNet, which lies in text-conditioned policy instantiation.

- **Empirical Insights into a New Paradigm.** We provide an extensive study across Mujoco and Meta-World benchmarks, highlighting both the promise and the limitations of language-enabled hypernetworks, and offering guidance for future extensions toward vision-grounded robotics.

## 2 RELATED WORK

**LLMs in Robotics.** Large language models (LLMs) have recently been integrated into robotics systems to enable natural language instruction following and high-level planning. Early efforts such as SayCan (Brohan et al., 2023) and PaLM-E (Driess et al., 2023) use pretrained LLMs to ground natural language into action primitives executed by low-level controllers. These approaches leverage LLMs' world knowledge but remain limited to symbolic or goal-level guidance.

Other works connect language and control indirectly. Code-as-Policies (Liang et al., 2023) translates instructions into robot API calls, Code-as-Rewards (Venuto et al., 2024) converts descriptions into reward signals, and SayTap (Tang et al., 2023) maps commands into foot contact patterns. These methods creatively bridge instruction and control, but depend on predefined APIs or accurate simulators, limiting real-world use. More recently, vision-language-action models such as RT-2 (Zitkovich et al., 2023), OpenVLA (Kim et al., 2025), and OCTO (Team et al., 2024) extend LLMs with visual grounding, but their scale and computational demands hinder deployment on resource-constrained robots.

In addition, recent work explores aligning language with behavior through contrastive representation learning. For example, CLASP (Rana et al., 2023) learns joint language–state–action embeddings and explicitly models the many-to-many correspondence between textual descriptions and demonstrations using distributional encoders. However, CLASP focuses on representation pretraining rather than generating executable policies. In TeNet, contrastive alignment plays a different and more limited role: we adopt standard contrastive objectives purely as an auxiliary mechanism to stabilize language-conditioned hypernetwork training, and we do not claim novelty at the level of the contrastive loss.

**Compact Sequence Models for Policy Learning.** In contrast to large LLM- or VLM-based systems, another line of research explores compact sequence models as policies for reinforcement learning. The Decision Transformer (DT) (Chen et al., 2021) recasts offline RL as a conditional sequence modeling problem, generating actions autoregressively given states and return-to-go. While effective in single-task settings, DT does not inherently support multi-task generalization, since it lacks a mechanism to distinguish tasks.

Several extensions introduce task-conditioning via trajectory prompts. Prompt-DT (Xu et al., 2022) improves adaptability by conditioning policies on a demonstration from the target task, and Meta-DT (Wang et al., 2024) extends this approach in a meta-learning setting. Although these methods improve transfer, they still require access to trajectory prompts at test time, which limits their practicality in real-world deployments where demonstrations are costly or unavailable. Diffusion-based models have also been explored for multi-task reinforcement learning in *state-based* domains, such as MTDiff (RL) (He et al., 2023) and MetaDiffuser (RL) (Ni et al., 2023), which condition on prompt trajectories or task-specific contexts to generalize across tasks. More recently, LPDT (Yang & Xu, 2024) aims to reduce data inefficiency by initializing Prompt-DT with a pre-trained language model and adding prompt regularization, but it still depends on trajectory prompts and yields mixed results across domains. DPDT (Zheng et al., 2024) tackles gradient conflicts in multi-task training by decomposing prompts into cross-task and task-specific components with test-time adaptation, yet it remains non–language-enabled and, without released code, its reproducibility is limited.

In parallel, modern *visuomotor* diffusion policies such as Diffusion Policy (Chi et al., 2025) use diffusion architectures to generate actions directly from images and have demonstrated strong real-world capabilities. These approaches differ fundamentally from the state-based RL methods discussed above. We focus on DT-based baselines to maintain architectural symmetry across all methods and because our experiments operate in low-dimensional state-based domains. Extending TeNet with diffusion-based trajectory encoders or diffusion-generated policy parameters is a promising direction for future work.

Overall, compact sequence models demonstrate that lightweight architectures can be applied to multi-task RL, but their reliance on trajectory prompts and lack of direct language grounding constrain their scalability as instruction-following agents.

**Hypernetworks and Meta-Learning.** Hypernetworks (Ha et al., 2016) generate the weights of another network and have been explored as a mechanism for rapid specialization in reinforcement learning. By conditioning on task-specific signals, a shared hypernetwork can instantiate new policies without retraining from scratch, making them attractive for meta-learning settings (Beck et al., 2023).

Recent works differ in their choice of conditioning signal:

- *Task-embedding based.* HyperZero (Rezaei-Shoshtari et al., 2023) enables zero-shot policy generation from structured task embeddings, while HyPoGen (Ren et al.) biases the generated weights for robust fine-tuning under distribution shift. A common alternative in multi-task RL is to condition policies on simple task identifiers such as one-hot vectors or learned task embeddings. While lightweight, these identifiers offer no semantic structure and cannot generalize to unseen tasks or continuous task families. Moreover, task IDs run counter to the goal of language-enabled policy generation, as they replace rich natural-language descriptions with opaque symbolic labels.

- *Trajectory-based.* Latent Weight Diffusion (Hegde et al., 2024) combines a diffusion model with a hypernetwork decoder to generate closed-loop policies from demonstrations. A related approach is Make-an-Agent (Liang et al., 2024), which conditions a diffusion model

on trajectory embeddings to synthesize policy weights. Unlike TeNet, these methods require demonstration trajectories at test time and therefore produce trajectory-conditioned policies rather than language-instantiated ones.

- *Archive-based.* Latent Policy Diffusion (LPD) (Hegde et al., 2023) distills a large QD-RL archive into a single diffusion model over policies, conditioned on behavior measures or short language labels. Unlike our work, which uses rich task descriptions as the primary conditioning signal, LPD relies on precomputed archives and uses text only as auxiliary behavior tags.

- *Morphology-based.* HyperDistill (Xiong et al., 2024) conditions a hypernetwork on robot morphology for embodiment transfer.

- *Image-based.* HUPA (Gklezakos et al., 2022) generates task-specific policies directly from image observations.

- *Language-based (outside robotics).* In NLP, hypernetworks have been used to generate adapter or LoRA weights directly from task descriptions or instructions, e.g., Hypter (Ye & Ren, 2021), HyperFormer (Mahabadi et al., 2021), HyperLoRA (Lv et al., 2024), and Text-to-LoRA (T2L) (Charakorn et al., 2025). These methods focus on adapting large language models, not synthesizing control policies.

These efforts show the versatility of hypernetworks for conditioning across modalities. However, existing works either rely on structured task descriptors, demonstrations, or morphology signals, or they use language only to adapt large models in NLP or vision. None directly combine LLM-based text encoders with hypernetworks to synthesize compact, task-specific robot control policies.

**Summary.** Prior work has explored LLM/VLM-based instruction following, compact transformer- and diffusion-based policies, and hypernetworks conditioned on tasks, trajectories, or morphology. Yet, no existing approach combines natural language grounding with hypernetwork-based policy synthesis. To our knowledge, our framework is the first to directly generate compact robot policies from language by aligning task descriptions with demonstrations and instantiating policies via a shared hypernetwork.

## 3 PROBLEM STATEMENT

**Language-Augmented MDP (LA-MDP).** We model a single task as a Language-Augmented MDP

$$\tilde{\mathcal{M}} = (\mathcal{S}, \mathcal{A}, P, R, \mu, H, \mathbb{L}), \tag{1}$$

which extends a standard MDP by including a language descriptor. The first six elements $(\mathcal{S}, \mathcal{A}, P, R, \mu, H)$ are the standard MDP components: $\mathcal{S}$ is the state space, $\mathcal{A}$ the action space, $P(s' \mid s, a)$ the transition dynamics, $R(s, a)$ the reward function, $\mu$ the initial state distribution, and $H$ the horizon. The additional component $\mathbb{L} \in \Delta(\mathcal{L})$ is a *language descriptor*, i.e., a probability distribution over natural-language strings in the space $\mathcal{L}$. Each task is associated with its own descriptor distribution $\mathbb{L}$, which generates natural-language paraphrases (e.g., "move forward" vs. "go straight") of the same underlying dynamics $P$ and reward function $R$. Thus, the LA-MDP can be viewed as a standard MDP augmented with a generative source of equivalent task descriptions. A policy $\pi(a \mid s)$ induces a trajectory distribution in $\tilde{\mathcal{M}}$, and its performance is

$$J(\pi) = \mathbb{E}\left[\sum_{t=0}^{H-1} R(s_t, a_t)\right], \tag{2}$$

with the task-optimal policy $\pi^* = \arg\max_{\pi \in \Pi} J(\pi)$.

**Multi-task LA-MDP.** We consider a distribution over tasks, where each task $\tau \in \mathcal{T}$ is an LA-MDP

$$\tilde{\mathcal{M}}_\tau = (\mathcal{S}_\tau, \mathcal{A}, P_\tau, R_\tau, \mu_\tau, H, \mathbb{L}_\tau). \tag{3}$$

Tasks may differ in $\mathcal{S}_\tau, P_\tau, R_\tau, \mu_\tau$ and $\mathbb{L}_\tau$, while sharing the action space $\mathcal{A}$. The multi-task objective is to learn a single policy that maximizes expected return across tasks: $\pi^* = \arg\max_{\pi \in \Pi} \mathbb{E}_{\tau \sim p(\mathcal{T})}\big[J_\tau(\pi)\big].$

**Offline setting.** No online interaction is permitted. The learner receives a static dataset collected from training tasks $\mathcal{T}_{\text{train}}$, each modeled as an LA-MDP

$$\mathcal{D}_{\text{train}} = \big\{ (\mathcal{X}_\tau, \mathcal{D}_\tau) \mid \tau \in \mathcal{T}_{\text{train}} \big\}, \tag{4}$$

where $\mathcal{X}_\tau = \{\xi_\tau^{(k)}\}_{k=1}^K$ is a set of expert trajectories $\xi_\tau^{(k)} = (s_0, a_0, r_0, \ldots, s_H)$, and $\mathcal{D}_\tau = \{d_\tau^{(m)}\}_{m=1}^M$ are i.i.d. descriptions sampled from the language descriptor, $d_\tau^{(m)} \sim \mathbb{L}_\tau$.

**Multi-task learning.** The learner is trained on demonstrations from a set of tasks $\mathcal{T}_{\text{train}}$. The objective is to learn a single model that approximates $\pi_\tau^*$ for all $\tau \in \mathcal{T}_{\text{train}}$, exploiting shared structure across tasks instead of training disjoint policies.

**Meta-learning.** The learner is trained on a collection of tasks $\mathcal{T}_{\text{train}}$ with the objective of generalizing to previously unseen tasks $\tau \in \mathcal{T}_{\text{test}}$. The challenge is to acquire transferable structure from $\mathcal{T}_{\text{train}}$ that enables rapid policy instantiation for new tasks without further environment interaction.

**Few-shot adaptation (baselines).** A common meta-RL strategy is to provide a small number of expert trajectories from the unseen task as adaptation data (few-shot setting). Prompt Decision Transformers (Prompt-DT) implement this by using short expert rollouts (*prompt trajectories*) as test-time task identifiers.

**Language-based instantiation (ours).** In contrast, we do not rely on prompt trajectories; instead we leverage natural-language descriptions sampled from $\mathbb{L}_\tau$ to instantiate policies for $\tau \in \mathcal{T}_{\text{test}}$, requiring the learner to ground language into behavior.

## 4 METHOD

### 4.1 OVERVIEW

Our framework, **TeNet (Text-to-Network)**, synthesizes compact, task-specific robot policies directly from natural language descriptions by conditioning a hypernetwork on language embeddings. At training time (Figure 1, top), the model receives task descriptions and expert demonstrations. Task descriptions are first encoded into text embeddings. Expert demonstrations supervise the policy through an imitation loss. In the grounded variant, we additionally introduce a trajectory encoder, and align its embeddings with the text embeddings (i.e., language grounding), thereby enriching the language representation with behavioral semantics. At inference time (Figure 1, bottom), a new task description is passed through the text encoder, projected to the appropriate embedding space, and fed into the hypernetwork to generate a policy that can be executed without further demonstrations.

We present two variants of our approach: **Direct TeNet**, which conditions the hypernetwork solely on text embeddings, and **Grounded TeNet**, which aligns text embeddings with trajectory embeddings during training to capture behavioral semantics and improve generalization.

### 4.2 DIRECT TENET

In the Direct TeNet variant, policies are instantiated directly from task descriptions without trajectory grounding. Given a description $d \in \mathcal{L}$, the text encoder $f_{\text{text}}$ produces an embedding $z_d = f_{\text{text}}(d) \in \mathbb{R}^{d_z}$. A projection network $g$ maps $z_d$ into the conditioning space of the hypernetwork: $\tilde{z}_d = g(z_d)$. The hypernetwork $h$ then generates the parameters $\theta_\pi$ of a task-specific policy network $\pi_{\theta_\pi}$

$$\theta_\pi = h(\tilde{z}_d), \qquad \pi_{\theta_\pi}(a \mid s). \tag{5}$$

Training relies on expert demonstrations $\xi_\tau = \{(s_t, a_t)\}_{t=0}^H$ from task $\tau$. The policy is supervised by behavior cloning (imitation learning)

$$\mathcal{L}_{\text{BC}} = -\mathbb{E}_{(s,a) \sim \xi_\tau} \big[ \log \pi_{\theta_\pi}(a \mid s) \big]. \tag{6}$$

Thus, Direct TeNet provides a simple mechanism for mapping language directly into executable policies through the hypernetwork.

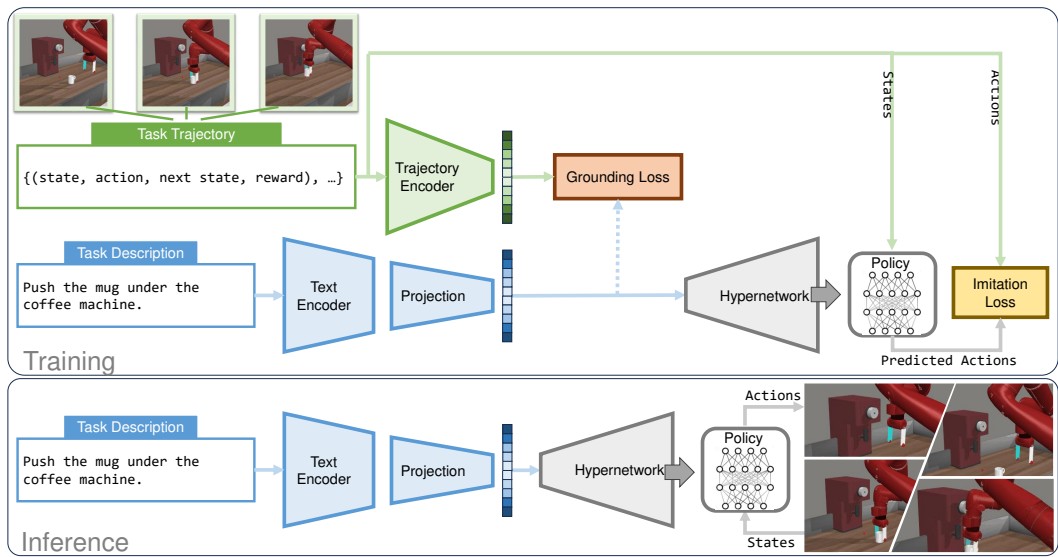

Figure 1: Training (top) and inference (bottom) of the proposed framework. During training, trajectories and task descriptions are encoded, projected, and aligned through a language grounding module, with a hypernetwork generating task-specific policies optimized by imitation and grounding losses. At inference, only the task description conditions the hypernetwork to instantiate a policy that maps states to actions.

## 4.3 GROUNDED TENET

Direct TeNet instantiates policies solely from projected text embeddings (Section 4.2). To better capture behavioral semantics, Grounded TeNet augments training with additional grounding objectives that align text and trajectory embeddings. We emphasize that grounding is not the primary conceptual contribution of TeNet: it is an auxiliary mechanism that stabilizes and enriches the text embeddings, while the core novelty lies in generating executable policy parameters directly from natural language.

Given an expert trajectory $\xi = \{(s_t, a_t, r_t, s_{t+1})\}_{t=0}^{H}$, the trajectory encoder $f_{\text{traj}}$ produces an embedding $z_\xi = f_{\text{traj}}(\xi)$. Both $z_\xi$ and the projected text embedding $\tilde{z}_d$ are mapped into a shared space, and a grounding loss $\mathcal{L}_{\text{ground}}$ is applied. We explore two variants:

**Direct alignment (MSE).** A simple strategy is to directly minimize the squared distance between projected text and trajectory embeddings

$$\mathcal{L}_{\text{align}} = \mathbb{E}_{(d,\xi)}\big[\|\tilde{z}_d - z_\xi\|_2^2\big]. \tag{7}$$

This objective enforces absolute closeness of paired embeddings in the shared space.

**Contrastive alignment.** Let $\text{sim}(\cdot, \cdot)$ denote cosine similarity and $\beta > 0$ a temperature parameter. For each update, we consider a finite candidate set of trajectory embeddings $\mathcal{C}_\xi$ and a finite candidate set of text embeddings $\mathcal{C}_d$ that provide negatives for the contrastive normalization.

*(i) Text–trajectory contrastive (symmetric).* For paired $(\tilde{z}_d, z_\xi)$, we align text to trajectory and trajectory to text with a symmetric InfoNCE

$$\mathcal{L}_{\text{text-traj}} = \tfrac{1}{2}\,\mathbb{E}_{(d,\xi)}\left[-\log \frac{\exp\big(\text{sim}(\tilde{z}_d, z_\xi)/\beta\big)}{\sum_{\xi' \in \mathcal{C}_\xi} \exp\big(\text{sim}(\tilde{z}_d, z_{\xi'})/\beta\big)} \;-\; \log \frac{\exp\big(\text{sim}(\tilde{z}_d, z_\xi)/\beta\big)}{\sum_{d' \in \mathcal{C}_d} \exp\big(\text{sim}(\tilde{z}_{d'}, z_\xi)/\beta\big)}\right]. \tag{8}$$

*(ii) Text–text contrastive.* Task descriptions can be structurally similar (e.g., differing only in goal parameters), which may collapse text embeddings. To encourage description-level discrimination,

we add

$$\mathcal{L}_{\text{text-text}} = \mathbb{E}_d \left[ - \log \frac{\exp\left(\text{sim}(\tilde{z}_d, \tilde{z}_d)/\beta\right)}{\sum_{d' \in \mathcal{C}_d} \exp\left(\text{sim}(\tilde{z}_d, \tilde{z}_{d'})/\beta\right)} \right]. \tag{9}$$

The final contrastive objective is $\mathcal{L}_{\text{contrastive}} = \mathcal{L}_{\text{text-traj}} + \mathcal{L}_{\text{text-text}}$.

**Summary.** The total training loss combines imitation learning with grounding: $\mathcal{L} = \mathcal{L}_{\text{BC}} + \lambda_{\text{g}} \mathcal{L}_{\text{ground}}$, where $\mathcal{L}_{\text{ground}}$ may include $\mathcal{L}_{\text{align}}$ or $\mathcal{L}_{\text{contrastive}}$, and $\lambda_{\text{g}}$ balances their contribution. At inference time, no trajectories are required – the policy is instantiated from text alone. Grounding is used only during training to shape the representation.

## 5 EXPERIMENTS

We conduct an extensive empirical study to evaluate TeNet and to provide insights into the design and behavior of language-enabled hypernetworks. Our experiments are performed on Mujoco control benchmarks (HalfCheetah-Vel, HalfCheetah-Dir, Ant-Dir) and Meta-World manipulation benchmarks (ML1 Pick-Place, MT10, MT50), covering both multi-task and meta-learning settings.

Beyond reporting standard performance, our goal is to systematically answer a series of questions about when and why TeNet is effective, how grounding influences policy quality, and how design choices such as hypernetwork structure, fine-tuning strategies, and task scaling affect performance. This section is therefore organized around these questions, with results interleaved with analysis.

### 5.1 EXPERIMENTAL SETUP

**Benchmarks.** We evaluate on *Mujoco* locomotion (HalfCheetah-Dir, HalfCheetah-Vel, Ant-Dir) and *Meta-World* manipulation (ML1 Pick-Place, MT10, MT50), spanning multi-task and meta-learning regimes. Full task definitions, state/action spaces, and splits are in App. A.

**Models.** We compare **DT** (Chen et al., 2021), **Prompt-DT** (Xu et al., 2022), and three TeNet variants: **TeNet** (direct, no grounding), **TeNet-MSE** (MSE grounding), and **TeNet-Contrast** (contrastive grounding). Implementation details, Prompt-DT size variants, and the Prompt-DT+Hypernetwork modification are in App. B.3.

**Metrics & protocol.** We report *episodic return* on Mujoco and *success rate* on Meta-World, plus *controller size* and *control frequency* for deployability. Metrics and definitions are in App. B.4. Results are averaged over 3 seeds; each task is evaluated with 50 rollouts (App. B.2).

**Defaults.** Unless stated otherwise: the text encoder is *Llama-3 8B* (frozen), the trajectory encoder is *Prompt-DT* (used only for grounded variants), and TeNet uses a small MLP hypernetwork to instantiate a $\sim$40K-parameter policy. Training is strictly offline. Architectural and optimization details are in App. B.1–B.2; system setup is in App. B.5.

### 5.2 RESULTS

Figure 2 summarizes performance across all six benchmarks, with a shared legend shown on top.

Several general trends are clear. First, **DT** is consistently the weakest model across all domains, confirming that a compact sequence model without explicit task signals is not suitable for multi-task or meta-learning. Both **Prompt-DT** and **TeNet** address this limitation by providing task signals, but they do so in fundamentally different ways: Prompt-DT relies on short expert rollouts (prompt trajectories) as identifiers, while TeNet derives task signals directly from natural language descriptions. This text-based conditioning avoids the need for demonstrations at test time, making TeNet more scalable and practical *within our state-based multi-task benchmarks*, as it removes the requirement for task-specific trajectory prompts.

Second, when comparing **TeNet** variants (more specifically **TeNet-Contrast**) against **Prompt-DT**, we observe consistent advantages. TeNet-Contrast outperforms Prompt-DT in HalfCheetah-Dir and

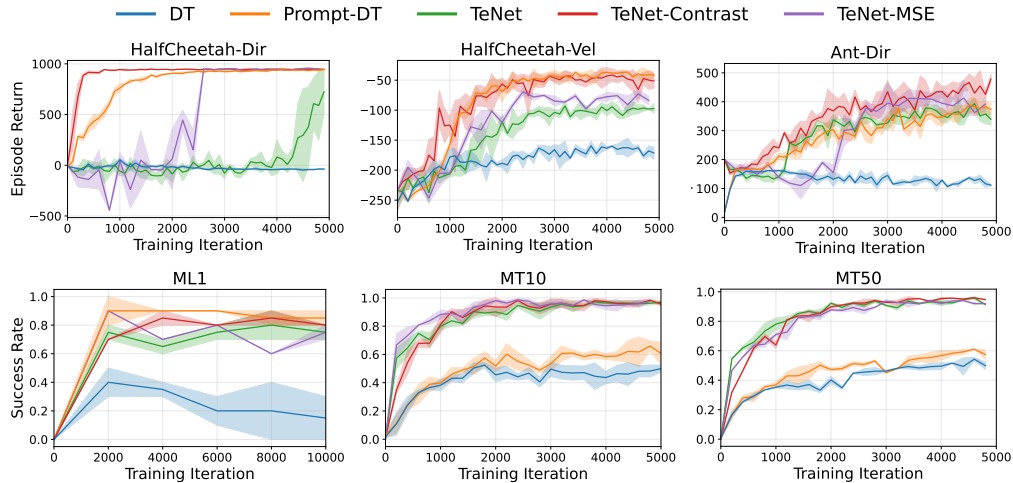

Figure 2: Performance across Mujoco (HalfCheetah-Dir, HalfCheetah-Vel, Ant-Dir) and Meta-World (ML1 Pick-Place, MT10, MT50). Each subplot reports mean and standard deviation over three seeds. A shared legend is shown at the top.

Ant-Dir, matches it in HalfCheetah-Vel, and is slightly worse in ML1 Pick-Place (which we analyze further in Section 5.7). Most strikingly, in MT10 and MT50 TeNet-Contrast *hugely outperforms* Prompt-DT. This large gap prompted us to investigate why Prompt-DT struggles so severely in multi-task benchmarks and to identify which design choices in TeNet are responsible for its robust performance. We return to this question in later subsections, where we dissect the role of task diversity, grounding, and hypernetwork conditioning.

### 5.3 CAN WE DIRECTLY BUILD POLICIES FROM LANGUAGE, OR DO WE NEED GROUNDING?

The results in Figure 2 reveal a mixed picture. Direct TeNet already provides a substantial improvement over DT across all benchmarks, confirming that natural language is an effective source of task signals. However, its relative performance compared to Prompt-DT depends critically on the setting. On **meta-learning benchmarks** (HalfCheetah-Vel, Ant-Dir, ML1 Pick-Place), Direct TeNet falls behind Prompt-DT, suggesting that text encodings, while informative, do not generalize to unseen tasks as effectively as trajectory prompts. In contrast, on **multi-task benchmarks** (MT10, MT50), Direct TeNet consistently outperforms Prompt-DT. These results indicate that *direct language-to-policy instantiation is viable* and scales well in diverse multi-task regimes, but that *additional grounding is required for robust generalization* in meta-learning settings where the agent must extrapolate to unseen tasks.

### 5.4 HOW SHOULD WE GROUND LANGUAGE IN BEHAVIOR?

The results in Figure 2 show that grounded TeNet, regardless of the chosen strategy, consistently outperforms Direct TeNet on the meta-learning benchmarks (HalfCheetah-Vel, Ant-Dir, ML1 Pick-Place). This confirms that additional grounding is necessary for robust generalization to unseen tasks.

Among the grounding methods, **contrastive alignment** generally performs better than direct alignment (MSE). The reason is that MSE enforces absolute closeness between paired text and trajectory embeddings, but provides no mechanism to separate embeddings from different tasks. As a result, embeddings from similar descriptions may collapse, limiting discriminability. In contrast, contrastive objectives simultaneously *pull together* matching text–trajectory pairs and *push apart* non-matching pairs, yielding a representation space that is both semantically aligned and better separated across tasks. This improved structure in the shared embedding space translates into stronger policy generalization.

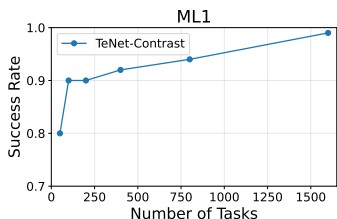

Figure 3: TeNet-Contrast performance on ML1 Pick-Place with varying numbers of tasks.

Table 1: Success rate on MT10 and MT50, along with controller size and control frequency. Prompt-DT-S is the default configuration.

| Model | Success Rate | | Ctrl Size | Ctrl Freq. |
| --- | --- | --- | --- | --- |
| | MT10 | MT50 | | |
| Prompt-DT-S | 0.73 | 0.61 | 1M | 557 Hz |
| Prompt-DT-M | 0.79 | 0.65 | 6M | 331 Hz |
| Prompt-DT-L | 0.74 | 0.58 | 39M | 190 Hz |
| Prompt-DT-HN | 0.99 | 0.97 | 5M | 462 Hz |
| TeNet | **0.99** | **0.98** | **40K** | **9300 Hz** |

### 5.5 Why does Prompt-DT struggle in MT10 and MT50?

The Meta-World multi-task benchmarks (MT10 and MT50) contain tasks that are far more distinct than those in Mujoco (e.g., pick-place versus drawer-open, compared to velocity or direction variations). This task diversity poses a major challenge for Prompt-DT. Furthermore, as the number of tasks increases, the success rate of Prompt-DT drops (from 0.73 on MT10 to 0.61 on MT50; see Figure 2). To better understand this gap, we conduct two follow-up experiments.

First, we ask whether the failure is simply due to *insufficient model capacity*. If trajectory prompts are expressive enough, then increasing the size of Prompt-DT (from small to medium to large) should yield meaningful improvements. Table 1 shows that this is not the case: larger Prompt-DT models achieve only marginal gains, indicating that the issue lies deeper than model capacity.

Second, we test whether the limitation arises from the lack of *task-specific parameterization*. In this variant, Prompt-DT-HN serves as a trajectory-conditioned hypernetwork baseline, where the prompt trajectory is encoded and used to generate policy weights via a shared hypernetwork. To this end, we add a hypernetwork on top of Prompt-DT to generate policy parameters conditioned on task signals. Table 1 indicates that this modification yields a substantial boost in success rates on both MT10 and MT50. The comparison demonstrates that explicitly generating task-specific parameters is crucial when dealing with distinct multi-task benchmarks. TeNet naturally benefits from this principle while also being language-enabled, removing the reliance on demonstration prompts.

### 5.6 How fast are TeNet policies?

Beyond task success, deployability depends critically on the efficiency of the policy: controllers must be compact enough to fit on resource-constrained robots, and fast enough to support high-frequency control loops. Table 1 reports both the number of parameters (controller size) and the control frequency that the method can sustain. For details on the computation of these two metrics, refer to App. B.4.

The results highlight a stark contrast. Prompt-DT variants range from 1M to 39M parameters, with control frequencies between 190 Hz and 600 Hz. Adding a hypernetwork further increases model size to 5M parameters, while improving task success, but the resulting policies remain limited to the sub-kHz regime. In contrast, TeNet policies contain only **40K parameters** and sustain control rates of over **9 kHz**, more than an order of magnitude faster than all Prompt-DT baselines.

These results demonstrate that TeNet not only matches or exceeds success rates but also provides *lightweight and high-frequency controllers*, making it well-suited for deployment on real robots where hardware constraints and responsiveness are critical.

### 5.7 Does scaling the number of training tasks improve TeNet's generalization?

In Section 5.2 we noted that TeNet-Contrast slightly underperforms Prompt-DT on ML1 Pick-Place. To investigate further, we study how scaling the number of training tasks affects generalization. Specifically, we vary the number of ML1 tasks available during training (50, 100, 200, 400, 800, 1600), while always holding out 10% of tasks for testing. The results are shown in Figure 3.

Performance improves steadily from a success rate of 0.80 with 50 tasks to 0.99 with 1600 tasks. This indicates that scaling the diversity of training tasks substantially enhances TeNet's ability to generalize. One possible factor is that as the number of training tasks grows, the domain gap between train and test tasks decreases, making generalization easier. In any case, reaching a success rate of **99%** with 1600 training tasks shows that TeNet can fully solve ML1 Pick-Place when provided with sufficient data. These results highlight both the promise and the data demands of language-enabled hypernetworks: like foundation models in other domains, TeNet benefits strongly from scale, even if it is data hungry.

### 5.8 Summary of Empirical Insights

Across benchmarks, we find that: (i) direct text-to-policy instantiation is viable, but grounding improves generalization; (ii) contrastive alignment provides stronger task discrimination than direct MSE alignment; (iii) hypernetworks enable task-specific parameterization that is critical for diverse multi-task benchmarks; (iv) TeNet policies are highly compact and sustain control frequencies above 9 kHz, far exceeding Prompt-DT baselines; and (v) scaling the number of training tasks substantially improves generalization, albeit at the cost of more data. Together, these findings establish TeNet as a compact and language-enabled alternative to trajectory-prompted models. We also compare LLaMA (Touvron et al., 2023) and BERT Devlin et al. (2019) text encoders under increasing paraphrasing complexity (App. C.5) and find that while both perform similarly on simple descriptions, LLaMA is substantially more robust to medium and hard paraphrases, leading to more stable policy instantiation.

In addition, we conduct ablation studies (App. C) to disentangle the contribution of individual components. These include isolating the role of the *text–text* contrastive term, assessing the effect of conditioning strategies during training, comparing frozen versus fine-tuned text encoders, and testing robustness to multiple natural-language descriptions of the same task. Together, these analyses reinforce the empirical claims of the main paper and clarify when TeNet is most effective.

## 6 Discussion

Our results provide evidence that compact, language-enabled hypernetworks can close much of the gap between lightweight sequence models and large VLAs within state-based, offline imitation settings. TeNet policies achieve strong performance while being orders of magnitude smaller and faster. However, the framework relies on high-quality task descriptions and currently focuses on imitation learning in simulation. These choices limit applicability to real robots and leave open the question of reinforcement fine-tuning and multimodal (vision + language) grounding. Deploying TeNet on real robots introduces additional challenges, including variability and noise in real-world trajectories (e.g., partial or inconsistent demonstrations) and the domain shift between simulation and physical dynamics. While the contrastive grounding objective is naturally robust to moderate noise, practical deployment will likely require collecting short expert demonstrations, handling visual perception, and potentially fine-tuning the instantiated policy with reinforcement learning. Addressing these limitations is an important direction for future work.

## 7 Conclusion

We presented TeNet, a text-to-network framework that instantiates compact, task-specific policies directly from natural language descriptions. By combining LLM embeddings, trajectory grounding, and hypernetwork-based parameter generation, TeNet produces lightweight controllers that generalize across tasks without requiring test-time demonstrations. Experiments on Mujoco and Meta-World benchmarks show that TeNet outperforms Prompt-DT in multi-task learning, achieves competitive meta-learning performance, and sustains control frequencies above 9 kHz. These findings establish language-enabled hypernetworks as a promising paradigm for scalable and deployable robot learning.

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

## LLM USAGE

We used large language models (LLMs) solely for writing assistance, including polishing, proof-reading, and minor sentence rewriting for clarity. LLMs were not involved in research ideation, experiment design, analysis, or any other substantive scientific contributions.

## A   BENCHMARKS

In Section 5, we briefly summarized the benchmarks to highlight the scope of our empirical study. Here we provide full specifications of all environments, including task definitions, state and action spaces, and train/test splits. Our evaluation covers two widely used families of continuous-control benchmarks: (i) *Mujoco control tasks* (Todorov et al., 2012), which probe multi-task learning and meta-learning in locomotion domains with goals such as direction or velocity, and (ii) *Meta-World manipulation tasks* (Yu et al., 2020), which test multi-skill generalization and large-scale multi-task policy synthesis in robotic manipulation. Together, these benchmarks span simple multi-task settings, meta-learning that requires generalization to unseen task specifications, and diverse manipulation skills, providing a comprehensive testbed for language-to-policy instantiation.

### A.1   MUJOCO CONTROL TASKS

We use three standard continuous-control benchmarks from the Mujoco physics engine (Todorov et al., 2012), following prior work in multi-task and meta-reinforcement learning (Xu et al., 2022). These tasks test whether policies instantiated from language can generalize across locomotion goals such as direction or velocity.

**HalfCheetah-Dir.**   This benchmark consists of two tasks: moving the half-cheetah agent either *forward* or *backward*. The state space has 20 dimensions (joint positions and velocities), and the action space has 6 dimensions (torque controls). Since there are only two tasks, the benchmark is treated as a *multi-task* setting: both tasks are included in training and evaluation.

**HalfCheetah-Vel.**   In this benchmark, tasks are defined by target forward velocities sampled uniformly from the interval $[0, 3]$. Each task specifies a different target velocity, and the reward encourages the agent to match this velocity. Following standard splits, we use 45 training tasks and 5 held-out test tasks. This benchmark is therefore a *meta-learning* setting, requiring the model to generalize to unseen velocity targets. The state and action spaces are the same as in HalfCheetah-Dir (20D states, 6D actions).

Figure 4: Representative Meta-World tasks used in our experiments, shown as sequences of frames (one task per row). From top to bottom: Coffee Push, Plate Slide Back Side, Push Wall, and Window Close. These examples illustrate the diversity of skills present in Meta-World.

**Ant-Dir.** This benchmark defines tasks by target locomotion directions sampled uniformly on the unit circle. Each task specifies a desired heading angle, and the reward encourages the ant agent to move in that direction. We use 45 training tasks and 5 held-out test tasks. The ant has a 27-dimensional state space (positions, velocities, contacts) and an 8-dimensional action space (joint torques). Like HalfCheetah-Vel, this is a *meta-learning* benchmark, since the agent must generalize to unseen movement directions at test time.

## A.2 META-WORLD MANIPULATION TASKS

We evaluate on the *Meta-World* benchmark suite (Yu et al., 2020), a standard collection of robotic manipulation tasks based on a simulated Sawyer robot arm. All Meta-World environments share a 39-dimensional state space (robot joint positions, object poses, etc.) and a 4-dimensional action space (3D end-effector displacements plus the gripper control). These tasks test both fine-grained skill variation and large-scale multi-task learning (See Figure 4).

**ML1 Pick-Place.** The ML1 benchmark focuses on variations of a single skill: picking up an object and placing it at a specified goal location. We adopt the pick-place environment, where tasks differ in the object–goal configuration. By default, we use 100 distinct tasks, with 90 used for training and 10 held out for testing. In scaling experiments, we vary the number of tasks to $\{50, 200, 400, 800, 1600\}$ while always holding out 10% for testing. This benchmark is a *meta-learning* setting, requiring generalization to novel goal configurations.

**MT10.** The MT10 benchmark consists of 10 distinct manipulation skills, such as pick-place, push, drawer-open, and shelf-place. For each skill, 50 goal configurations are randomly sampled and fixed, resulting in a total of 500 training tasks. Since all tasks are included in training, this is a *multi-task* benchmark, and evaluation is performed on the same set of tasks.

**MT50.** The MT50 benchmark extends MT10 to 50 distinct manipulation skills, again with 50 random goal configurations per skill. This yields a total of 2,500 training tasks, covering a broad range of manipulation behaviors including object placement, pushing, pulling, opening/closing, and container manipulation. As in MT10, this is a *multi-task* benchmark: training and evaluation cover the same 50 skills and their associated goal distributions.

## A.3 SUMMARY OF TASK PROPERTIES

Table 2 summarizes the key properties of all benchmarks used in our experiments, including the dimensionality of state and action spaces, the number of tasks and splits, and whether the setting is multi-task or meta-learning.

| Benchmark | State Dim. | Action Dim. | #Tasks (Train / Test) | Setting |
|---|---|---|---|---|
| HalfCheetah-Dir | 20 | 6 | 2 (2 / 0) | Multi-task |
| HalfCheetah-Vel | 20 | 6 | 50 (45 / 5) | Meta-learning |
| Ant-Dir | 27 | 8 | 50 (45 / 5) | Meta-learning |
| ML1 Pick-Place | 39 | 4 | 100 (90 / 10)† | Meta-learning |
| MT10 | 39 | 4 | 500 (500 / 0) | Multi-task |
| MT50 | 39 | 4 | 2500 (2500 / 0) | Multi-task |

Table 2: Summary of benchmark properties. All Meta-World environments share a 39D state space and 4D action space. †In scaling experiments, the number of ML1 tasks is varied between 50 and 1600 while always holding out 10% for testing.

### A.4 EXAMPLE TASK INSTRUCTIONS

To make the language inputs concrete, we provide representative natural-language instructions used by TeNet across all benchmarks:

- **HalfCheetah-Vel:** "Move forward with target velocity 2.0 m/s."
- **Ant-Dir:** "Walk in the direction of 125 degrees."
- **ML1 Pick-Place:** "Pick up the block and place it at position $(-0.1, 0.2, 0.1)$."
- **Meta-World MT (examples):**
  - "Open the sliding door."
  - "Pull the drawer open."
  - "Close the drawer."
  - "Press the top-down button."
  - "Insert the peg into the side hole."
  - "Push the block to the right side."

These examples illustrate the range of language instructions used throughout the experiments and help contextualize TeNet's text-conditioned policy instantiation.

## B    IMPLEMENTATION DETAILS

In Section 5, we provided only a high-level overview of the experimental setup to remain within the page limit. Here, we include the complete implementation details of our framework, covering the architecture of each component, the training procedure, and the system configuration. This appendix is intended to support reproducibility and to clarify design choices that are only briefly mentioned in the main paper.

### B.1    MODEL ARCHITECTURE

Our framework consists of a text encoder, a trajectory encoder (for grounded variants), a projection network, a hypernetwork, and a policy network. Below we describe each component in detail.

**Text encoder.**    We use the pretrained *LLaMA-3 8B* model (Touvron et al., 2023) to encode natural language task descriptions. The text encoder is invoked only once per task at policy instantiation; its output conditions the hypernetwork, and it is never used inside the control loop. Consequently, the encoder size has no effect on control frequency or runtime performance. Unless otherwise stated, the encoder is kept *frozen* during training to preserve general-purpose language representations. In ablation studies (Appendix C), we also evaluate LoRA-based fine-tuning of the text encoder, but find that it reduces performance in low-data regimes. Smaller encoders such as BERT Devlin et al. (2019) can also be used without affecting runtime performance; however, as shown in Appendix C.5, larger encoders offer greater robustness to paraphrastic variation in natural-language task descriptions.

**Trajectory encoder.** For grounded variants of TeNet, we employ a *Prompt Decision Transformer* (Prompt-DT) (Xu et al., 2022) as the trajectory encoder. Given an expert demonstration, the encoder produces a trajectory embedding that captures behavioral semantics of the task. We removed the action prediction head and use the final hidden representation as an embedding. In addition, we set the embedding dimension to 256 (instead of the default 128) so that it matches the projected text embeddings. This trajectory embedding is then used both for alignment with text embeddings and, in the Grounded-Flow variant, as an additional conditioning input to the hypernetwork. For direct TeNet, this component is omitted. The trajectory encoder is trained jointly with the rest of the TeNet architecture: it receives gradients from the imitation loss and, for grounded variants, from the grounding objectives. There is no separate pretraining stage; the encoder is optimized end-to-end together with the projection head and hypernetwork.

**Projection network.** The text embedding is passed through a two-layer MLP with ReLU activation to be projected into a conditioning space of dimension 256. This projection ensures that both modalities are comparable and suitable for conditioning the hypernetwork. We denote this module as $g(\cdot)$ in the main text.

**Hypernetwork.** The hypernetwork $h(\cdot)$ is a two-hidden-layer MLP with 128 units per layer and ReLU activations. Its output is a multi-head vector that parameterizes the weights of each layer of the downstream policy network. For example, one head produces the weight matrix for the first policy layer, another produces the bias vector, and so on. This design ensures modular generation of policy parameters while keeping the hypernetwork compact.

**Policy network.** The instantiated policy $\pi_{\theta_\pi}$ is a two-hidden-layer MLP with 128 units per layer and ReLU activations. The input is the state vector of the environment (encoded as a 128-dimensional vector using a linear transformation), and the output is an action distribution over the continuous control space. For Mujoco tasks, the action dimension is 6 (HalfCheetah) or 8 (Ant), while for Meta-World tasks it is 4. This network contains only $\sim$ 40K parameters, making it lightweight and suitable for high-frequency control.

### B.2 TRAINING SETUP

All experiments are conducted in the *offline* setting: models are trained exclusively from expert demonstrations without additional environment interaction. We summarize the training procedure here; formal definitions of the loss functions are provided in Section 4.

**Loss functions.** All models are trained with a behavior cloning objective on expert trajectories. Grounded variants additionally use the alignment objectives introduced in Section 4, namely mean-squared alignment, contrastive alignment, and the text–text contrastive term (for TeNet-Contrast). The overall loss is a weighted sum of imitation and grounding terms, with $\lambda_g$ controlling the relative contribution of grounding.

**Grounded-Flow mechanism.** We study a dual-path variant (*Grounded-Flow*) in which, during training, we run two forward passes through the shared hypernetwork – one conditioned on text embeddings and one on trajectory embeddings — apply imitation losses to both, and backpropagate their (weighted) sum. At inference, only the text-conditioned path is retained. Figure 6 shows that removing this dual-path supervision reduces performance on ML1 Pick-Place.

**Optimization.** Following the setup of Prompt-DT (Xu et al., 2022), we use the AdamW optimizer with a learning rate of $1 \times 10^{-4}$ and weight decay of $1 \times 10^{-4}$. A linear warm-up schedule is applied for the first 10k steps, implemented with a `LambdaLR` scheduler in PyTorch:

$$\eta_t = \min\left\{\tfrac{t+1}{10000},\, 1\right\} \eta_0,$$

where $\eta_t$ is the effective learning rate at step $t$ and $\eta_0$ the base rate. Gradient norms are clipped at 0.25 using `torch.nn.utils.clip_grad_norm_`. Batch sizes are set per benchmark: 32 for Mujoco, 32 for ML1, 10 for MT10, and 50 for MT50. Training runs for 5k iterations on Mujoco, MT10 and MT50, and 10k iterations on ML1. Unless otherwise noted, the text encoder is frozen and only the projection head, hypernetwork, and policy are updated. At inference time, the text encoder

is not executed: the policy parameters are generated once from the encoded description, and action selection depends solely on the low-dimensional state input.

**Task descriptions.** By default, we use a single natural-language description per task during training and evaluation. In Appendix C, we show that TeNet is insensitive to the number of descriptions: adding multiple paraphrases per task does not significantly affect performance.

**Evaluation protocol.** All reported results are averaged over three independent runs with different random seeds. For each task, we evaluate over 50 rollouts and report the mean and standard deviation across tasks and seeds. In multi-task benchmarks (HalfCheetah-Dir, MT10, and MT50), evaluation is performed on the training tasks, whereas in meta-learning benchmarks (HalfCheetah-Vel, Ant-Dir, and ML1 Pick-Place), evaluation is performed on the held-out test tasks.

### B.3 MODELS COMPARED

We evaluate TeNet against established compact sequence models and several of its own variants. Below we summarize all models considered.

**Decision Transformer (DT).** The Decision Transformer (Chen et al., 2021) is a representative compact sequence model that formulates reinforcement learning as conditional sequence modeling. We re-implement DT following the original paper, using the same hidden dimension and number of layers, and apply it to the offline multi-task datasets. Since DT does not include task-conditioning, it serves as a lower-bound baseline.

**Prompt Decision Transformer (Prompt-DT).** Prompt-DT (Xu et al., 2022) extends DT to the few-shot setting by conditioning policies on short expert rollouts (prompt trajectories) at test time. We adopt the default architecture and optimization setup from the original paper, ensuring a fair comparison to our method. Prompt-DT is included both as a trajectory encoder within TeNet and as a standalone baseline. Unlike TeNet, it requires access to demonstration prompts at inference.

**Prompt-DT size variants.** To test whether limited capacity explains Prompt-DT's performance gap, we implemented three model sizes: *small* (default), *medium*, and *large*. The presets are as follows:

- **Small (default):** 3 layers, embedding dimension 128, 1 head (head dimension 128), inner dimension 512, ReLU activation, dropout 0.1.

- **Medium:** 6 layers, embedding dimension 256, 4 heads (head dimension 64), inner dimension 1024, ReLU activation, dropout 0.1.

- **Large:** 12 layers, embedding dimension 512, 8 heads (head dimension 64), inner dimension 2048, ReLU activation, dropout 0.1.

These follow the scaling rules of transformer architectures. As shown in Table 1, increasing size yields only marginal gains, indicating that lack of capacity is not the main bottleneck.

**Prompt-DT with hypernetwork (Prompt-DT-HN).** To test whether task-specific parameterization improves performance, we modify Prompt-DT by removing its action prediction head and replacing it with a hypernetwork. The hypernetwork generates the parameters of the downstream policy conditioned on task signals. This variant achieves a substantial boost on MT10 and MT50 (Table 1), demonstrating the importance of task-specific parameterization. Prompt-DT-HN acts as a trajectory-conditioned hypernetwork baseline: the Prompt-DT trajectory encoder produces a task embedding, which conditions a hypernetwork that generates the full policy parameters.

**TeNet.** Our text-to-network model instantiates policies directly from natural-language task descriptions, without any grounding objectives (Section 4.2). It demonstrates the viability of language-based policy generation even in the absence of trajectory alignment.

**TeNet-MSE.** A grounded variant of TeNet that employs direct mean-squared-error alignment between text and trajectory embeddings (Section 4.3). This variant tests whether simple embedding closeness is sufficient for grounding.

**TeNet-Contrast.** Our strongest grounded variant, which uses contrastive alignment objectives to align text and trajectory embeddings while preserving task discriminability (Section 4.3). This variant consistently provides the best generalization performance across benchmarks.

**Additional variants.** In ablation studies, we also evaluate further modifications of TeNet, such as alternative grounding strategies, text encoder fine-tuning, and Grounded-Flow conditioning. These results are reported in Appendix C.

### B.4 METRICS

We report both task-level performance metrics and system-level efficiency metrics.

**Episodic return (Mujoco).** For Mujoco benchmarks, performance is measured by the average episodic return over evaluation rollouts. Returns are reported in the raw reward scale of the environment (no normalization).

**Success rate (Meta-World).** For Meta-World benchmarks, performance is measured by the success rate, defined as the fraction of evaluation rollouts in which the environment signals task completion (e.g., object placed at target, drawer fully opened).

**Controller size.** To assess deployability, we report the number of parameters of the controller used at inference time. For DT and Prompt-DT, this equals the full model size, since the transformer is executed online at every step. For TeNet, the hypernetwork and encoders are used only once at policy instantiation; at inference, only the generated policy network is executed. Thus, the reported controller size for TeNet corresponds to the instantiated policy parameters ($\sim$40K), reflecting the actual runtime footprint.

**Control frequency.** We also report the everage action generation rate (Hz) sustained by each model on a single NVIDIA GPU. For DT and Prompt-DT, this reflects the inference speed of the entire transformer model, which typically operates in the sub-kHz regime. In contrast, TeNet executes only the compact instantiated policy at inference, while the hypernetwork and encoders are used once at instantiation time. As a result, TeNet policies sustain control rates above 9 kHz, more than an order of magnitude faster than DT-based baselines.

Control frequency in Table 1 was measured by timing repeated calls to the policy inside the evaluation loop. Specifically, we warmed up the model with 50 calls and then measured the average latency over 500 calls at step 20 of an evaluation episode, using `time.perf_counter()` and explicit CUDA synchronization. This benchmark excludes environment stepping and I/O, and therefore reflects policy-only inference speed.

### B.5 SYSTEM SETUP

All experiments were run on a workstation equipped with an AMD Ryzen Threadripper PRO 5975WX CPU (32 cores, 64 threads), 128 GB of RAM, and a single NVIDIA RTX A6000 GPU (48 GB memory). Training a single TeNet model typically required between 6–12 hours depending on the benchmark (shorter for Mujoco, longer for Meta-World MT50). Control frequency measurements (Table 1) were obtained on the same hardware.

We use PyTorch together with HuggingFace Transformers for the text encoder and PEFT for LoRA-based fine-tuning. Meta-World and Mujoco environments are taken from their official open-source implementations. Random seeds are fixed across runs for reproducibility. All reported results are averaged over three seeds as described in Appendix B.2. **The complete source code will be released publicly after the reviewing process is completed.**

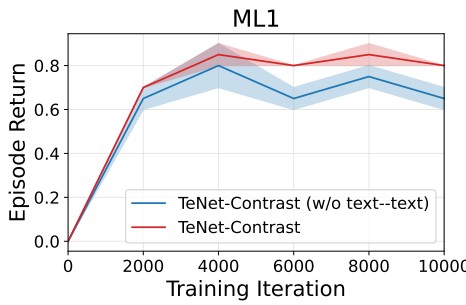 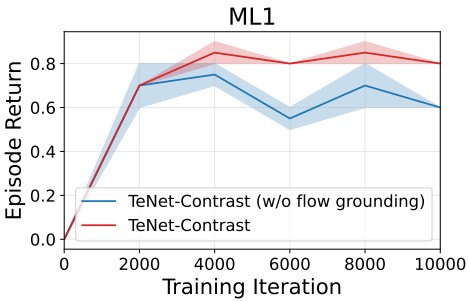

Figure 5: **Ablation on the text–text contrastive term.** Performance on ML1 Pick-Place with and without the text-text contrastive term.

Figure 6: **Ablation on Grounded-Flow.** Performance on ML1 Pick-Place with and without Grounded-Flow.

## C    ABLATION STUDIES

To better understand the design choices underlying TeNet, we conducted a series of ablation studies. These experiments isolate the contribution of different components and training strategies, allowing us to assess their individual impact on generalization and performance. Specifically, we examine (i) the contribution of the *text–text* component of the contrastive objective (toggling this term while keeping the text–trajectory term active), (ii) the effect of *Grounded-Flow*, where trajectory embeddings additionally condition the hypernetwork, (iii) the influence of fine-tuning the pretrained text encoder compared to keeping it frozen, and (iv) the robustness of TeNet to multiple natural-language descriptions of the same task. Together, these studies provide a deeper understanding of when and why TeNet is effective, and they reinforce the empirical claims presented in the main paper.

### C.1    EFFECT OF THE TEXT–TEXT CONTRASTIVE TERM

We ablate the contribution of the *text–text* component of the contrastive objective in TeNet-Contrast, comparing the full model against a variant we denote as **TeNet-Contrast (w/o text–text)**. This ablation is conducted on ML1 Pick-Place, where descriptions differ only in the target coordinates, e.g., ``Pick the object and place it at (tx, ty, tz).'' Such descriptions are lexically very similar, which makes the corresponding text embeddings prone to collapse into overlapping clusters. As shown in Figure 5, removing the text–text term reduces success rates, indicating that it plays a critical role in maintaining discriminability among task descriptions. The justification is that the text–text contrastive term explicitly pushes apart embeddings from different tasks, preventing collapse and ensuring that policies conditioned on these embeddings generalize more effectively.

### C.2    EFFECT OF GROUNDED-FLOW

We study the effect of *Grounded-Flow*, where trajectory embeddings are used not only for alignment but also to condition the hypernetwork alongside text embeddings during training. This design allows gradients from the imitation loss to propagate through both pathways, so that policy parameters are shaped jointly by text and trajectory signals. At inference, however, only text embeddings are available, and the policy is instantiated exactly as in the standard model.

Figure 6 shows that removing Grounded-Flow reduces performance on ML1 Pick-Place. Although trajectories are still used for alignment in this ablation, they no longer contribute direct conditioning during training. The likely explanation is that Grounded-Flow acts as an auxiliary channel that strengthens the training signal: trajectory embeddings encode rich task dynamics, and conditioning the hypernetwork on them forces the parameter space to better capture the correspondence between text and behavior. As a result, when only text is available at inference, the model is more effective at instantiating the correct policy.

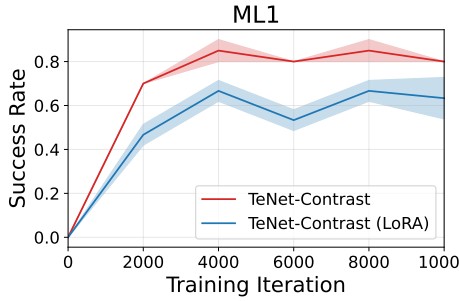

Figure 7: **Ablation on fine-tuning the text encoder.** Performance on ML1 Pick-Place with a frozen encoder versus LoRA fine-tuning.

Table 3: **Effect of multiple task descriptions.** Success rates of TeNet-Contrast on MT10 and MT50 when varying the number of task descriptions per task. Minor variations are due to random seed effects.

| | Success Rate | | | | |
|---|---|---|---|---|---|
| | # Task Descriptions | | | | Avg. |
| | 1 | 2 | 5 | 10 | |
| MT10 | 0.99 | 0.98 | 0.98 | 0.98 | 0.98 |
| MT50 | 0.98 | 0.99 | 0.98 | 0.98 | 0.98 |

### C.3 EFFECT OF FINE-TUNING

By default, TeNet freezes the parameters of the pretrained text encoder and only trains the projection, hypernetwork, and policy components. In this ablation, we instead apply LoRA (Hu et al., 2021) to fine-tune the text encoder and evaluate the effect on ML1 Pick-Place.

Figure 7 shows that LoRA fine-tuning leads to substantially worse performance compared to the frozen encoder. The likely reason is data scarcity: ML1 contains only 100 tasks in total (90 for training), and the corresponding descriptions are highly similar, differing mainly in goal coordinates. Under such conditions, LoRA fine-tuning tends to overfit to the limited training descriptions, reducing the ability of the encoder to generalize to unseen tasks. In contrast, keeping the encoder frozen preserves its broader linguistic representations, resulting in stronger downstream performance.

We note, however, that these results may not be fully conclusive: we have not systematically studied different LoRA parameter configurations. Reducing the number of additional learnable parameters may mitigate overfitting and yield different outcomes, which we leave for future investigation.

### C.4 EFFECT OF MULTIPLE TASK DESCRIPTIONS

A key strength of TeNet is that it conditions policies on natural language rather than fixed task identifiers. This enables flexible interaction: users can provide different descriptions of the same task, and the model can still instantiate the correct policy. By contrast, prior approaches that rely on task IDs cannot accommodate such variability. In practice, large language encoders (e.g., LLaMA) map paraphrases with the same intent to nearby embeddings, allowing TeNet to treat multiple descriptions consistently. For example, the Meta-World `pick-place-v3` task can be described in many different but equivalent ways:

```
"Pick up the object and place it at the target."
"Lift the item and move it to the goal."
"Carry the object to the designated location."
"Transport the item to the target spot."
"Grab the object and set it at the goal."
```

By default, we use a single task description per task during training and inference. In this ablation, we vary the number of task descriptions (1, 2, 5, 10) generated via a language model and evaluate TeNet-Contrast on MT10 and MT50. Table 3 reports the results. The set of paraphrases used in this ablation spans a broad range of lexical and syntactic variation, including multi-clause descriptions, narrative-style prompts, and additional contextual modifiers. Despite this diversity, Table 3 shows that TeNet remains stable across 1, 2, 5, or 10 paraphrases per task, indicating that its text encoder produces consistent embeddings for semantically equivalent instructions.

The results confirm that TeNet is insensitive to the number of task descriptions: success rates remain essentially unchanged whether a task is described once or with several paraphrases. Minor numerical differences are due to randomness across training seeds, not to the number of descriptions. This

insensitivity is expected, since modern LLM encoders produce similar embeddings for descriptions that express the same intent. Thus, TeNet can naturally support flexible human interaction without requiring carefully standardized task identifiers.

## C.5 ENCODER CHOICE AND PARAPHRASING ROBUSTNESS

We compare TeNet using LLaMA (Touvron et al., 2023) and BERT Devlin et al. (2019) on MT10 under increasing levels of paraphrastic complexity. For each task, we generate *10* Level 0 descriptions ("Easy") using a language model, and train all models on these 10 canonical paraphrases. At evaluation time, we provide *10* Level 1 paraphrases ("Medium") and *10* Level 2 paraphrases ("Hard") per task, allowing us to test the robustness of the text encoder under more complex linguistic variation. This ablation isolates how reliably the hypernetwork instantiation process behaves when the same task is described using increasingly unconstrained natural language.

**Paraphrasing Levels.** We show one representative example for each difficulty level:

- **Level 0 (Easy).** Short, canonical phrasing with minimal syntactic variation. Example: *"Reach the target position."*

- **Level 1 (Medium).** Longer descriptions containing additional clauses or modifiers. Example: *"Bring the end effector all the way to the target location without interacting with any objects."*

- **Level 2 (Hard).** Narrative-style phrasing with redundant wording or mild distractors. Example: *"Your goal is simply to drive the end effector toward the marked target point and stop exactly when you arrive at that location."*

The text encoder is invoked only once per task at policy instantiation. Therefore, robustness in this experiment reflects the encoder's ability to map semantically equivalent but lexically different descriptions to consistent embeddings.

**Results.**

Table 4: Success rates on MT10 when training on 10 Easy paraphrases and evaluating on 10 Medium or 10 Hard paraphrases.

| Encoder | Level 0 | Level 1 | Level 2 |
|---------|---------|---------|---------|
| **LLaMA** | 0.99 | 0.95 | 0.89 |
| **BERT** | 0.99 | 0.89 | 0.82 |

Both encoders achieve identical performance on Level 0 descriptions, showing that TeNet can reliably instantiate policies from simple instructions. However, as linguistic complexity increases, LLaMA proves substantially more robust: under Level 2 narrative-style paraphrases, LLaMA retains high performance while BERT suffers a significant drop. This indicates that richer language models create more stable embedding spaces for semantically equivalent instructions.

Figure 8 illustrates the same trend across training iterations. When trained on Easy paraphrases, LLaMA maintains stable and high success even when evaluated on harder paraphrases, whereas BERT shows clear degradation as linguistic complexity increases.

Overall, these results justify our choice of LLaMA as the default text encoder. Although the encoder is used only once per task and does not affect control frequency, its ability to produce stable embeddings under paraphrastic variation significantly improves TeNet's robustness to real-world instruction variability.

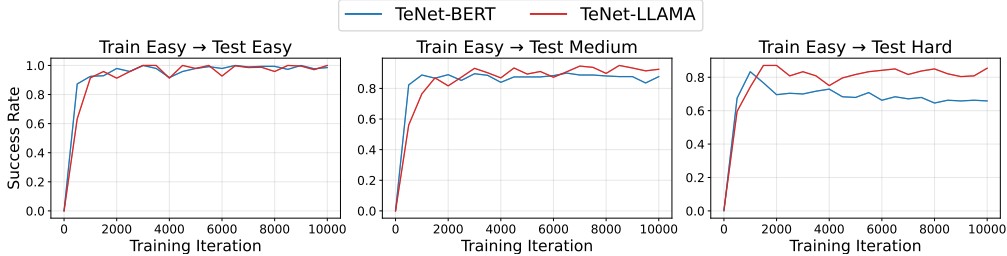

Figure 8: Training curves comparing TeNet with LLaMA and BERT text encoders. Models are trained on 10 Easy paraphrases and evaluated on Easy, Medium, and Hard paraphrasing levels. LLaMA maintains higher stability, especially under medium and hard paraphrastic variation.

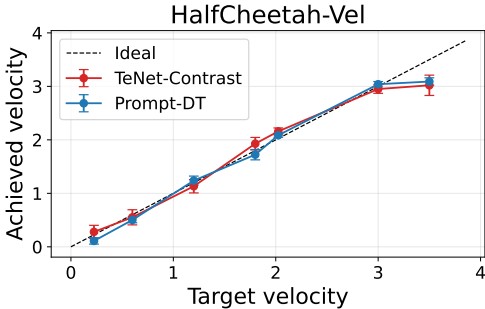

Figure 9: Achieved forward velocity as a function of instructed target velocity in HalfCheetah-Vel. Points show mean achieved speed over 50 rollouts for each instruction. TeNet-Contrast closely follows the target velocities on the held-out meta-test tasks and saturates near the environment's practical speed limit for an out-of-range instruction at $3.5$ m/s.

### C.6 VELOCITY-FOLLOWING BEHAVIOR IN HALFCHEETAH-VEL

The HalfCheetah-Vel benchmark is designed to evaluate velocity-tracking behavior. At each step, the reward is given by

$$r = - |v_{\text{current}} - v_{\text{target}}| , \tag{10}$$

so that the episodic return directly reflects how accurately the policy matches the commanded forward speed. This formulation is standard and used throughout prior work on this benchmark Xu et al. (2022).

The task distribution is constructed by defining target forward velocities on a fixed grid ranging from $0.075$ m/s to $3.0$ m/s, with uniform increments of $0.075$ m/s. From this grid, a subset of velocities is used for training, and a disjoint subset is reserved as held-out evaluation tasks. In our experiments, the unseen meta-test velocities are

$$0.225, \ 0.6, \ 1.2, \ 1.8, \ 2.025 \text{ m/s},$$

which are drawn from this grid but never seen during training.

To make TeNet's instruction-following behavior more explicit, we instantiate policies from natural-language commands of the form

"Move forward with target velocity $X$ m/s."

for each of the unseen evaluation velocities $X \in \{0.225, 0.6, 1.2, 1.8, 2.025\}$. In addition, we probe extrapolation beyond the benchmark's range by evaluating an out-of-distribution instruction with $X = 3.5$ m/s.

For each instruction, we execute the instantiated policy for $50$ rollouts and measure the average forward velocity. The achieved velocities are computed as the average forward speed over the last $20$

steps of each rollout. Figure 9 plots the achieved velocity as a function of the instructed target velocity for both TeNet-Contrast and Prompt-DT. Across all unseen evaluation velocities, TeNet-Contrast closely tracks the commanded speeds, indicating smooth generalization over the continuous family of velocity-tracking tasks. For the extrapolated instruction at 3.5 m/s, both TeNet-Contrast and Prompt-DT saturate near the upper end of the HalfCheetah dynamics (around $\sim 3$ m/s), reflecting the practical locomotion limit of the environment rather than a failure of instruction following.

## C.7 DISCUSSION

These ablations clarify the role of each design choice in TeNet. The text–text contrastive term proves important in benchmarks such as ML1, where task descriptions differ only minimally, by preventing embedding collapse and preserving task discriminability. Grounded-Flow further improves training by allowing trajectory-conditioned gradients to shape the hypernetwork, leading to stronger policies even though inference remains text-only. In contrast, fine-tuning the text encoder with LoRA harms performance in the low-data regime of ML1, highlighting that frozen language encoders provide more robust generalization when only limited descriptions are available. Finally, the multiple-description study confirms that TeNet is insensitive to paraphrasing and description multiplicity, underscoring the practical advantage of language-based conditioning over task identifiers.

