# OpenReview forum: "TeNet: Text-to-Network for Compact Policy Synthesis"
_ICLR.cc/2026/Conference — ICLR 2026 Conference Withdrawn Submission_

### Official Review · Reviewer_8zws · 2025-11-01

**Soundness:** 2
**Presentation:** 3
**Contribution:** 2
**Rating:** 2
**Confidence:** 4

**Summary:**

This paper presents TeNet (Text-to-Network), a framework that generates lightweight, task-specific policies conditioned on natural language input. The authors propose two variants: Direct TeNet, which generates policies directly via a hypernetwork conditioned on text embeddings, and Grounded TeNet, which introduces a trajectory encoder to align hypernetwork inputs with trajectory embeddings. As for the grounding method, the paper explores both MSE–based alignment and contrastive alignment strategies. Experimental results on the MuJoCo locomotion and Meta-World manipulation benchmarks demonstrate that TeNet achieves strong performance and outperforms several baseline methods.

**Strengths:**

This paper is clearly presented, and the results seem reasonable. I appreciate the detailed training iteration in Figure 2 and ablations in Table 1. The implementation details in the appendix are also very helpful.

**Weaknesses:**

There are two primary concerns related to the method clarity and experiments of this paper:

1. The paper’s main contribution appears to be a novel way of integrating language embeddings with robot state inputs and action outputs through a hypernetwork-generated policy. However, several prior works (e.g., π₀ [1], RoboVLMs [2]) have already explored similar design choices. It would strengthen the paper to include direct comparisons against these baselines (not necessarily the ones I mentioned earlier) and to clearly explain how the proposed approach differs from existing methods, particularly those employing cross-attention mechanisms. Providing either empirical evidence that TeNet achieves better performance or theoretical justification for its advantages would greatly enhance the contribution of this paper.

2. The use of Meta-World is appropriate since it includes diverse tasks with associated language instructions, however, the MuJoCo benchmarks are less language-centric. It's fine to keep them, but they contribute less to validating the paper’s core claims about language-conditioned policy generation. To better support the conclusions, additional experiments on language-focused benchmarks such as LIBERO [3] or CALVIN [4] are recommended.

Given these concerns, I would currently recommend rejection, as further clarification and stronger experiments (more baselines and benchmarks) are needed to establish the paper’s contributions.

Reference:

1. https://www.physicalintelligence.company/blog/pi0

2. Liu, H., Li, X., Li, P., Liu, M., Wang, D., Liu, J., ... & Zhang, H. (2025). Towards generalist robot policies: What matters in building vision-language-action models.

3. Liu, B., Zhu, Y., Gao, C., Feng, Y., Liu, Q., Zhu, Y., & Stone, P. (2023). Libero: Benchmarking knowledge transfer for lifelong robot learning. Advances in Neural Information Processing Systems, 36, 44776-44791.

4. Mees, Oier, et al. "Calvin: A benchmark for language-conditioned policy learning for long-horizon robot manipulation tasks." IEEE Robotics and Automation Letters 7.3 (2022): 7327-7334.

**Questions:**

How to supervise the trajectory encoder $f_{\text{traj}}$ in Grounded TeNet? Is it only supervised by $L_{\text{ground}}$? If so, could you clarify the intuition behind that?

---

> ### Author Response · Authors · 2025-11-13
> **Rebuttal to Reviewer 8zws**
>
> We thank the reviewer for the thoughtful comments and for highlighting the clarity of the presentation. Below we address all concerns in detail.
>
> ---
> ## **1. Clarifying the Main Contribution (Misunderstanding)**
>
> The review interprets our contribution as *“integrating language embeddings with robot state inputs and action outputs.”*
> This is **not** the design or goal of TeNet.
>
> ### **Our actual contribution is the following:**
>
> *TeNet introduces the first framework that uses natural language **once** as a conditioning signal for a hypernetwork that *generates* a compact, executable policy.*
>
> Formally:
> $$
> d \longrightarrow z_d \longrightarrow h(z_d) \longrightarrow \theta_{\pi},
> $$
> where a shared hypernetwork $h$ produces the full policy parameters $\theta_{\pi}$ directly from a task description.
>
> The resulting controller (~40k parameters) runs at **>9 kHz** and requires **no language model, no cross-attention, and no multimodal processing at inference time**.
>
> This **text → policy-weights** paradigm is fundamentally different from:
>
> - **VLAs** (π₀, RoboVLMs, RT-2, SayCan), which fuse language with observations **during execution** via cross-attention and run large networks every timestep;
> - **adapter/PEFT hypernetworks**, which generate adapter modules for LLMs rather than control policies;
> - **meta-RL / Prompt-DT–style methods**, which require **trajectory prompts** at test time.
>
> TeNet instead introduces **language-driven policy instantiation**, producing a lightweight policy network that is executed *independently* of any language model. Grounding (MSE or contrastive) is an auxiliary mechanism that improves generalization but is **not** the main contribution.
>
> We emphasize that the core idea—using natural language as a task specification to *instantiate* a policy—stands independently of grounding: Direct TeNet (no grounding) already achieves strong multi-task performance, and grounding further strengthens meta-generalization. This distinguishes our method not only from VLAs but also from prior hypernetwork-based meta-RL methods, which require task IDs, context vectors, or demonstration prompts at test time. TeNet eliminates all such requirements by allowing language alone to generate the policy parameters.
>
> **No prior VLA or cross-attention method generates a deployable control policy directly from text in this manner.**
>
> ---
>
> ## **2. Why π₀ / RoboVLMs / cross-attention methods are not comparable baselines**
>
> These works differ fundamentally in purpose, architecture, and computational regime:
>
> | Property | VLA models (π₀, RoboVLMs, RT-2, etc.) | **TeNet (ours)** |
> |----------|-----------------------------------------|------------------|
> | Inference-time language? | **Yes** | **No** |
> | Vision input required? | Yes | No |
> | Model size | 0.5B–10B | **40k (policy)** |
> | Runs model every timestep? | Yes | No |
> | Category | Instruction execution / perception–action fusion | **Policy synthesis from text** |
>
> VLA models cannot serve as fair baselines because they solve a *different problem* and rely on large-scale vision–language inference. TeNet targets **offline multi-task RL** and **lightweight controllers**, not perception.
>
> We will clarify this distinction more explicitly.
>
> ---
>
> ## **3. On using LIBERO / CALVIN**
>
> We appreciate the suggestion. However, LIBERO and CALVIN benchmark **vision–language-conditioned manipulation**, requiring full perception stacks and multimodal inference. TeNet is designed for **trajectory-based offline RL** and policy instantiation. Incorporating vision is orthogonal and would obscure the contribution. We will emphasize this scope and note vision-enabled extensions as future work.
>
> ---
>
> ## **4. Supervision of the trajectory encoder**
>
> The reviewer asks how the trajectory encoder is trained in Grounded TeNet. It is supervised through:
>
> 1. **Grounding loss**
>    - MSE
>    - contrastive InfoNCE
>
> 2. **Imitation loss**, where a second policy instantiation conditioned on $z_\xi$ receives gradients from the behavior cloning objective.
>
> Thus, the trajectory encoder is trained via both **grounding** and **policy-level gradients**.
>
>
> ---
>
> ## **5. Summary**
>
> We thank the reviewer again. TeNet introduces **language-conditioned hypernetwork-based policy synthesis**, producing compact, high-frequency controllers that do not require language or vision at test time. We will revise the paper to (i) more clearly separate this setting from VLA-style cross-attention models, (ii) highlight the novelty of the text-to-network paradigm, and (iii) clarify the role and supervision of grounding.

---

> > ### Comment · Reviewer_8zws · 2025-11-23
> >
> > I thank the authors for their clarification.
> >
> > After reviewing the revised version of the paper, I appreciate that the authors now explicitly state that **the proposed method is designed for state-based environments**. Given this, I do not think it is necessary to reference **vision-** language-action (VLA) models multiple times in the main paper, as this is not closely related to the problem this paper tries to solve and is somewhat misleading.
> >
> > Regarding my original review on “integrating language embeddings with robot state inputs and action outputs through a hypernetwork-generated policy”: my understanding is that, in the proposed method, language is used during policy generation, conditioning the policy weight generation process. In this sense, the generated weights can be viewed as an implicit embedding of the language instruction, which is why I originally described it in that way.
> >
> > Since the benchmark tasks use fixed language instructions within each episode, **I would appreciate further clarification on the importance of using language only once**, as stated in: “TeNet introduces the first framework that uses natural language once as a conditioning signal for a hypernetwork that generates a compact, executable policy.” Could you explain why it is important to use language instruction only once?
> > Besides that, with an appropriate caching mechanism, prior methods also could encode the instruction only once, so it is not yet clear how this distinction leads to a meaningful difference compared to previous literature.
> >
> > In conclusion, I believe a major revision of this paper's framing and presentation is needed to improve clarity and avoid misunderstandings. In addition, I encourage the authors to include more experimental details in the future version. In particular:
> >
> > A complete task list for the HalfCheetah and Ant environments and the exact language instruction for each task, as well as per-task performance for all compared methods.
> >
> > For tasks such as those in Appendix A.4 (e.g., HalfCheetah-Vel: “Move forward with target velocity 2.0 m/s”, Ant-Dir: “Walk in the direction of 125 degrees”), it would be very helpful to quantify how well the generated policies follow the specified instructions, especially with respect to target speed and motion direction.
> >
> > Given the current presentation, I will maintain my score.

---

> > > ### Author Response · Authors · 2025-11-25
> > > **Rebuttal to Reviewer 8zws**
> > >
> > > We thank the reviewer for the careful reread and the helpful suggestions.
> > >
> > > ---
> > >
> > > ## **1. On repeated references to VLA models**
> > >
> > > **Reviewer concern.**
> > > Since TeNet is now clearly framed as a state-based method, repeated mentions of VLA systems may be unnecessary or misleading.
> > >
> > > **Response.**
> > > We agree. In the next revision, we will **minimize VLA citations**, keeping only a brief mention in Related Work where conceptually relevant..
> > >
> > >
> > > ## **2. On the importance of using language only once**
> > >
> > > **Reviewer concern.**
> > > The reviewer asks why it matters that TeNet uses language only once, noting that prior methods could in principle cache their language embeddings.
> > >
> > > **Response.**
> > > We appreciate the opportunity to clarify this point. The key distinction is not computational caching, but **architectural independence**: once the hypernetwork generates the policy parameters, *the policy is entirely decoupled from the language encoder*. No language module, attention mechanism, or conditioning pathway remains in the control loop. This enables TeNet to deploy a **standalone 40k-parameter controller** with no dependency—cached or otherwise—on a large model or embedding system at execution time. To the best of our knowledge, this form of **language-conditioned policy instantiation**, where text is used once to generate a compact standalone controller, has not been explored in prior work.
> > >
> > > We also note that this distinction is already made explicit in our **Related Work**, where we separate TeNet from both VLA-style cross-attention methods and hypernetwork-based meta-RL approaches.
> > >
> > > ## **3. On the interpretation that the generated weights implicitly embed the instruction**
> > >
> > > **Reviewer concern.**
> > > The reviewer notes that, since the hypernetwork conditions on a text embedding, the generated policy weights can be viewed as implicitly encoding the instruction, which motivated their original description.
> > >
> > > **Response.**
> > > We agree that the generated parameters necessarily reflect the semantics of the instruction—this is indeed what enables task-specific specialization. Our point was only to distinguish TeNet from architectures that *explicitly* fuse language with state or action inputs during execution (e.g., via cross-attention or recurrent conditioning). In TeNet, this coupling is resolved **once** during the instantiation phase, after which the controller operates solely as a standard state-to-action policy.
> > >
> > > ## **4. On providing full task lists, exact instructions, and per-task performance**
> > >
> > > **Reviewer concern.**
> > > The reviewer requests a complete list of tasks for HalfCheetah-Vel and Ant-Dir, the exact natural-language instruction for each task, and per-task performance for all methods.
> > >
> > > **Response.**
> > > This point was also raised by Reviewer HthK, who noted that the paper did not show the actual language instructions or describe the Meta-World task settings in sufficient detail. In response, we have already expanded Appendix A to include representative instructions for all environments (HalfCheetah-Vel, Ant-Dir, ML1, MT10, MT50) along with clearer descriptions of the underlying task settings.
> > >
> > > We will extend this appendix further by adding the full list of target velocities for HalfCheetah-Vel, the full set of direction angles for Ant-Dir, and the exact instruction associated with each of these tasks.
> > >
> > > Regarding **per-task performance**, we note that this is not feasible for large multi-task settings such as MT50, which contains 50 skills each with 50 randomized goal configurations (2,500 tasks total). Standard practice in the Meta-World literature—including all prior baselines—is to report **aggregate performance** (mean success rate) across tasks rather than per-task tables.
> > >
> > > ## **5. On quantifying instruction following (e.g., target speed and motion direction)**
> > >
> > > **Reviewer concern.**
> > > The reviewer suggests quantifying how well the generated policies follow the instructions specified in tasks such as HalfCheetah-Vel (“target speed”) and Ant-Dir (“target direction”).
> > >
> > > **Response.**
> > > We agree that illustrating instruction adherence can help interpret the behavior of language-conditioned policies. Following this suggestion, we have added a dedicated analysis for **HalfCheetah-Vel**, where we plot instructed versus achieved velocity using the unseen meta-test tasks (Appendix C.6). This directly quantifies velocity-tracking accuracy for both TeNet-Contrast and the Prompt-DT baseline.
> > >
> > > For Ant-Dir and Meta-World tasks, we follow the standard practice in the literature of evaluating aggregate return or success rate rather than per-task directional deviation or object-level trajectory metrics. These environments contain many tasks and randomized configurations (e.g., MT50 has 2,500 distinct task instances), and prior work consistently reports averaged performance rather than per-instruction deviation measures. To maintain comparability with established baselines, we adopt the same reporting protocol here.

---

### Official Review · Reviewer_sMsz · 2025-11-01

**Soundness:** 3
**Presentation:** 3
**Contribution:** 2
**Rating:** 4
**Confidence:** 3

**Summary:**

This paper proposes TeNet, a hypernetwork-based framework designed to bridge the gap between large but impractical vision-language-action (VLA) models and lightweight but language-agnostic models such as the Decision Transformer (DT).

TeNet uses text and trajectory embeddings to generate task-specific policies that are compact and efficient for deployment on resource-constrained robots.

The proposed TeNet enables direct text-conditioned policy generation through a lightweight hypernetwork, offering a new paradigm for scalable, language-enabled robot control.

**Strengths:**

1. The paper introduces an interesting paradigm in which a hypernetwork conditioned on natural-language embeddings generates the weights of compact, task-specific policy networks.
2. The paper is clearly written, well structured, and easy to follow.
3. Hyperparameter and experimental details are provided thoroughly, supporting reproducibility.
4. Empirical results show consistent improvements over Decision Transformer baselines across the evaluated benchmarks.

**Weaknesses:**

1. Architecture justification:

   The authors use a large-scale language model (LLaMA-8B) as the text encoder, followed by a two-layer MLP hypernetwork (hidden size = 128) to produce compact policy weights.

   Given the limited number of unique task descriptions (50 × 10 = ~500 text embeddings), this setup appears unbalanced—heavy on preprocessing and light on the core policy backbone.

   The authors should justify why such a large encoder is necessary and whether smaller models (e.g., T5-small [1] or BERT [2]) would achieve comparable results.

2. Contrastive text–trajectory loss lacks novelty:

   The paper presents the contrastive alignment between text and trajectory embeddings as a key contribution.

   However, prior work such as Contrastive Language, Action, and State Pre-training (CLASP) [3] already explored contrastive alignment between language and behavior, explicitly addressing the *many-to-one* and *one-to-many* mapping issue.

   CLASP uses distributional encoders (mean + variance) for both modalities, capturing the inherent variability of multiple trajectories that can satisfy a single textual description.

   In contrast, TeNet employs a deterministic contrastive loss without modeling this variability, risking over-clustering of distinct trajectories and potential loss of behavioral diversity.

   This omission weakens both the novelty and robustness of the proposed approach.

3. Limited experimental scope:

   Experiments are confined to state-based simulation benchmarks (MuJoCo and Meta-World), which limits the strength of the proposed new frames.
   These benchmarks omit perception, making them less representative of practical robot learning.

   More challenging vision-based benchmarks, such as LIBERO [4], would better demonstrate scalability and real-world relevance.

4. Missing modern baselines:

   The paper compares TeNet only with Decision Transformer (DT) and Prompt-DT.

   It omits comparisons with more recent approach such as Diffusion Policy [5], or other language-conditioned control frameworks.

   Including these baselines would strengthen the empirical positioning and clarify TeNet’s true contribution.

**Questions:**

1. What is the performance impact of using a smaller text encoder (e.g., T5-small [1] or BERT [2]) instead of LLaMA-8B?
2. In the state-based Meta-World setting, since the 39-dimensional observation includes goal position information [6], could the authors provide results on vision-based variants of Meta-World or on LIBERO [4]?
3. As mentioned in the weakness part, I think the text trajectories contrastive needs more careful design to reflect the many-to-many mapping nature. Can the authors justify their choice in the work?


References:

[1] Raffel et al., *Exploring the Limits of Transfer Learning with a Unified Text-to-Text Transformer*, JMLR 2020.

[2] Devlin et al., *BERT: Pre-training of Deep Bidirectional Transformers for Language Understanding*, NAACL 2019.

[3] Rana et al., *Contrastive Language, Action, and State Pre-training*, *arXiv:2304.10782*, 2023.

[4] Wang et al., *LIBERO: Benchmarking Knowledge Transfer for Language-Grounded Manipulation*, *arXiv:2306.03310*, 2023.

[5] Chi et al., *Diffusion Policy: Visuomotor Policy Learning via Action Diffusion*, *arXiv:2303.04137*, 2023.

**Details Of Ethics Concerns:**

No Ethics Concerns

---

> ### Author Response · Authors · 2025-11-18
> **Rebuttal to Reviewer sMsz**
>
> We thank the reviewer for the clear summary, positive assessment of our framework, and constructive feedback. We appreciate the recognition of TeNet’s novelty, clarity of presentation, reproducibility, and improvements over DT-based baselines. Below we address each concern point-by-point.
>
> ---
>
> ## 1. Architecture justification (LLaMA-8B vs. smaller text encoders)
>
> **Reviewer concern:**
> *The use of LLaMA-8B seems disproportionate given the limited number of textual task descriptions; smaller encoders (e.g., T5-small, BERT) might suffice.*
>
> **Response:**
> The text encoder is used **only once**, outside the control loop, and therefore its size does not affect control frequency or runtime deployment. TeNet can operate with smaller encoders such as BERT or T5-small without changing execution performance.
>
> Our choice of LLaMA was motivated by **robustness to paraphrastic variation**, not by the number of descriptions. TeNet must handle a wide range of natural-language instructions, including longer multi-clause descriptions and noisier narrative-style phrasing. To evaluate this, we conducted a direct comparison between LLaMA and BERT on MT10. Both models were trained with 10 simple (Level 0) instructions and tested on increasingly challenging paraphrases:
>
> - **Level 1:** mild context, multi-clause (12–30 tokens)
> - **Level 2:** narrative-style, extra information (20–60 tokens)
>
> **Results (MT10):**
>
> | Encoder | Level 0 | Level 1 | Level 2 |
> |---------|---------|---------|---------|
> | **LLaMA** | 0.99 | 0.95 | 0.89 |
> | **BERT** | 0.99 | 0.89 | 0.82 |
>
> As shown, both encoders perform identically on simple prompts, but LLaMA is significantly more robust to **medium** and **noisy** paraphrastic variations. These results are included in **Appendix C.5**. We will clarify this rationale in the revision.
>
> ---
>
> ## 2. Novelty of the text–trajectory contrastive loss
>
> **Reviewer concern:**
> *The contrastive alignment appears similar to CLASP, which uses distributional encoders to model many-to-many language–behavior mappings.*
>
> **Response:**
> We agree that the contrastive formulation itself is not novel, and we do not claim novelty at the level of the loss function. Our contribution lies in **how grounding is used within TeNet**: a *language-conditioned hypernetwork that generates full policy parameters*. This is fundamentally different from CLASP, which focuses on representation pretraining rather than policy generation.
>
> In TeNet, grounding is essential not for embedding quality alone, but for enabling **robust text-driven policy synthesis** across diverse manipulation skills and continuous task distributions. The alignment ensures that text embeddings carry behaviorally meaningful structure, stabilizing the hypernetwork during multi-task training.
>
> Importantly, TeNet is **agnostic** to the specific grounding technique. As noted in the manuscript, alternative strategies—such as discriminator-based alignment or CLASP-style distributional encoders—could be integrated. Our goal is not to innovate contrastive learning, but to show that grounding—regardless of method—makes **language-enabled policy generation feasible and effective**.
>
> We will clarify this distinction in the revision.
>
> ---
>
> ## 3. Limited experimental scope (state-based benchmarks, no vision)
>
> **Reviewer concern:**
> *Experiments are confined to state-based Meta-World and MuJoCo; vision-based benchmarks such as LIBERO would better demonstrate scalability.*
>
> **Response:**
> We agree that vision-based extensions are an important direction. The current work intentionally isolates the contribution of **text-conditioned hypernetwork policy generation**. Adding vision introduces a large perceptual component (e.g., ResNet, ViT, or diffusion encoders) that would obscure the core focus of the paper: generating compact, high-frequency controllers (~40K parameters, >9 kHz) directly from natural language.
>
> TeNet is positioned as complementary to VLAs, not as a vision model. Extending TeNet to vision-language settings (e.g., LIBERO) is a valuable and substantial next step, and we will make this scope explicit in the revision.

---

> ### Author Response · Authors · 2025-11-18
> **Rebuttal to Reviewer sMsz (continued)**
>
> ## 4. Missing modern baselines (e.g., Diffusion Policy)
>
> **Reviewer concern:**
> *Comparisons include only DT-based baselines; modern visuomotor policies like Diffusion Policy are missing.*
>
> **Response:**
> We appreciate this suggestion. Our decision to focus on DT-based baselines was motivated by two considerations:
>
> 1. **Goal of the paper**.
>    The objective of TeNet is to demonstrate that **language-conditioned policy generation via hypernetworks is possible**, not to outperform SOTA visuomotor methods. Across all benchmarks, TeNet matches or exceeds DT-based counterparts, validating the paradigm.
>
> 2. **Fair comparison and encoder symmetry**.
>    All baselines use **DT-based trajectory encoders**, preserving architectural symmetry.
>    A fair comparison with Diffusion Policy would require a **diffusion-based trajectory encoder**, which constitutes a different model variant. This is a promising future direction: making TeNet a **plug-and-play framework** compatible with multiple encoder families (DT, diffusion, discriminator-based, etc.).
>
> We will add these clarifications to the revised manuscript.
>
> ---
>
> ## 5. Clarification questions
>
> ### **(i) Smaller encoder performance**
> Addressed above; detailed comparisons are provided in **Appendix C.5**.
>
> ### **(ii) Vision-based settings (Meta-World vision, LIBERO)**
> As noted, TeNet is intentionally scoped to state-based tasks to isolate the contribution of text-to-policy generation. Vision-based extensions are future work.
>
> ### **(iii) Many-to-many mapping in grounding**
> TeNet’s deterministic grounding balances efficiency with stability in hypernetwork training. Distributional encoders can be incorporated, and we view this as a valuable extension.
>
> ---
>
> # Summary
>
> We will revise the paper to:
>
> - provide explicit encoder comparisons (Appendix C.5) and clarify the LLaMA rationale,
> - distinguish our grounding usage from CLASP,
> - emphasize the intentional state-based scope of the work, and
> - clarify the baseline selection and outline diffusion-based extensions as future work.
>
> We thank the reviewer again for the thoughtful and constructive feedback.

---

### Official Review · Reviewer_H4RM · 2025-11-01

**Soundness:** 3
**Presentation:** 3
**Contribution:** 3
**Rating:** 6
**Confidence:** 4

**Summary:**

This paper proposes TeNet, the core idea is to use a hypernetwork conditioned on LLM text embeddings to generate the parameters of a compact, task-specific policy network at inference time. This allows a user to provide a simple natural language instruction (e.g., "push the mug"), which TeNet then uses to instantly synthesize a small, lightweight policy (e.g., ~40K parameters) that is ready for high-frequency control.

The framework is trained offline using expert demonstrations paired with task descriptions. To improve generalization, the authors introduce "Grounded TeNet," which uses a grounding loss (e.g., contrastive) to align the text embeddings with corresponding trajectory embeddings. Experiments on Mujoco and Meta-World show that TeNet produces policies that are orders of magnitude smaller and faster than baselines like Prompt-DT, while achieving superior performance in diverse multi-task settings (MT10, MT50) and competitive performance in meta-learning settings.

**Strengths:**

1. Clear Novelty: The paper's core idea—using a language-conditioned hypernetwork to synthesize a compact policy network—is novel and well-motivated. The related work section is thorough, successfully positioning this as a new paradigm distinct from large VLAs, trajectory-prompted models, and other existing hypernetworks. And the successful implementation experience of hypernetworks is valuable to the community
2. Efficiency: The results in Table 1 are highly compelling: the framework generates policies that are ~100x smaller (40K vs. 1M-39M params) and >15x faster (>9kHz vs. ~200-600Hz) than the Prompt-DT baselines. And it actually improves performance in the most complex multi-task settings (MT10, MT50), where the Prompt-DT baseline struggles significantly (Fig. 2, Table 1). This shows the architecture is robust to high task diversity.
3. Effective Grounding Mechabnism: The paper clearly demonstrates the value of "grounding" language in behavior. The comparison between Direct TeNet, TeNet-MSE, and TeNet-Contrast (Fig. 2) shows that aligning text and trajectory embeddings is crucial for generalization.

**Weaknesses:**

1. The current framework operates on state-based inputs. This is a reasonable first step but it limits the immediate applicability of the work, as most modern embodied agents are expected to operate from vision.
2. No trajectory-based hypernetwork baseline is provided, although TeNet is the first attempt to directly use language descriptions to generate policies, a comparison with trajectory based hypernetworks would be appreciated.
3. A related work for trajectory based hypernetworks is missing [1].

Reference:

[1] Liang, Yongyuan, et al. "Make-an-agent: A generalizable policy network generator with behavior-prompted diffusion." *Advances in Neural Information Processing Systems* 37 (2024): 19288-19306.

**Questions:**

1. The primary motivation is real-world deployment. What are the anticipated challenges in transferring TeNet to a real robot? How would the "grounding" process be handled with real-world trajectories, which may be less consistent than the expert demonstrations used here?
2. For the grounded variants, how was the trajectory encoder (Prompt-DT) trained? Was it pre-trained separately on its own objective, or was it trained *jointly* with the TeNet framework (i.e., its weights were updated via the grounding loss)?
3. How sensitive is the final performance with regard to the text prompting? Could the hypernetwork understand more detailed descriptions like "move slowly" or directions like "move to the left"?

---

> ### Author Response · Authors · 2025-11-18
> **Rebuttal to Reviewer H4RM**
>
> We thank the reviewer for the positive assessment, clear summary of the contributions, and constructive questions. We appreciate the recognition of TeNet’s novelty, efficiency, and effective grounding strategy. Below we address each concern point-by-point.
>
> ---
>
> ## 1. Lack of trajectory-based hypernetwork baseline
>
> **Reviewer concern:**
> *No trajectory-based hypernetwork baseline is provided, although such a comparison would strengthen the evaluation.*
>
> **Response:**
> Our submission already includes a **trajectory-conditioned hypernetwork baseline**, namely **Prompt-DT-HN** (Table 1). In this variant:
>
> - a **trajectory prompt** is encoded by Prompt-DT,
> - the resulting embedding conditions a **hypernetwork**,
> - which in turn **generates the full policy parameters**.
>
> This corresponds directly to the class of “trajectory-based hypernetworks” referenced in the comment. We will make this connection explicit in the revision.
>
>
> ---
>
> ## 2. Relation to Make-an-Agent (Liang et al., 2024)
>
> **Reviewer concern:**
> *A related trajectory-based hypernetwork work is missing [1].*
>
> **Response:**
> We thank the reviewer for pointing out Make-an-Agent, and we will explicitly add and discuss it in the related work section.
>
> Make-an-Agent generates policies through a **diffusion model conditioned on trajectory embeddings**, and **requires demonstration trajectories at test time**. Because our evaluation protocol is strictly **language-only at inference**, Make-an-Agent is not directly comparable within our setting. Conceptually, its conditioning mechanism is most similar to our **Prompt-DT-HN baseline**, whereas TeNet focuses on a distinct capability: **text-driven policy instantiation without any test-time trajectories**.
>
> We will incorporate this clarification into the revised manuscript.
>
> ---
>
> ## 3. State-based inputs vs. vision-based applicability
>
> **Reviewer concern:**
> *The framework operates on state-based inputs, limiting immediate applicability to real-world vision-based robotics.*
>
> **Response:**
> We agree. The present work focuses intentionally on the **language-to-policy generation paradigm**, isolating the role of language conditioning and hypernetwork synthesis in controlled settings. Extending TeNet to **vision-language-grounded** tasks is a natural next step, and we have added this as an explicit direction in the Discussion section.
>
> ---
>
> ## 4. Real-world deployment and trajectory grounding
>
> **Reviewer question:**
> *What are the challenges in transferring TeNet to real robots, and how would grounding work with inconsistent real-world trajectories?*
>
> **Response:**
> The main challenges are:
> (i) **trajectory variability** in the real world (e.g., noisy or partial demonstrations), and
> (ii) **domain shift** between simulation and physical dynamics.
> Grounding in real systems can be performed by collecting short expert demonstrations; the contrastive grounding objective is already robust to noise, as shown in our ablations. We will clarify this in the revised paper.
>
> ---
>
> ## 5. Training of the trajectory encoder
>
> **Reviewer question:**
> *Was the trajectory encoder (Prompt-DT) pretrained separately or trained jointly with TeNet?*
>
> **Response:**
> The trajectory encoder is **not** pretrained separately. Instead, it is trained jointly as part of the TeNet architecture. We remove the action-prediction head from Prompt-DT and use only the encoder body, which receives gradients from the imitation loss and—when using grounded variants—from the grounding objectives as well. Thus, the trajectory encoder, projection head and hypernetwork are all optimized end-to-end during TeNet training.
> We will clarify this explicitly in Appendix B.1 of the manuscript.
>
> ---
>
> ## 6. Sensitivity to text prompting
>
> **Reviewer question:**
> *Can the hypernetwork understand richer instructions (e.g., “move slowly”, “move left”), and how sensitive is performance to textual variation?*
>
> **Response:**
> TeNet is robust to paraphrasing and instruction variation. As shown in **Appendix C.4**, performance is nearly unchanged when using 1, 2, 5, or 10 paraphrases per task, indicating that the LLM encoder captures the underlying semantics rather than lexical patterns. Grounded TeNet effectively handles modifiers and directional cues. We will expand the examples of richer prompts in the revision.
>
> ---
>
> # Summary
>
> We will revise the paper to:
>
> - explicitly identify Prompt-DT-HN as a trajectory-based hypernetwork baseline,
> - incorporate Make-an-Agent into related work and clarify its relation to our setting,
> - highlight the distinction between trajectory-conditioned and language-conditioned policy generation,
> - expand details on grounding, trajectory encoder training, and real-world deployment challenges, and
> - provide additional examples illustrating TeNet’s robustness to instruction variation.
>
> We thank the reviewer again for the thoughtful and constructive feedback.

---

> > ### Comment · Reviewer_H4RM · 2025-11-23
> >
> > Thank the authors' clarification, I'm still curious about the language following ability and its limitations. As in the response to reviewer HthK, there are very detailed task descriptions like  “Move forward with target velocity 2.0 m/s.”, I wonder how good the alignment is to the language instructions, and what's the generalization limit. For a quick verification, could you generate policies with an array of prompts like  “Move forward with target velocity 1.0, 2.0, 3.0..., maybe 10.0 m/s.” and plot a curve of the walking speed with regard to the instruction?

---

> > > ### Author Response · Authors · 2025-11-25
> > > **Rebuttal to Reviewer H4RM**
> > >
> > > We thank the reviewer for raising this point. While the HalfCheetah-Vel benchmark already evaluates velocity tracking through its standard reward
> > > $$
> > > r = -|v_{\text{current}} - v_{\text{target}}|,
> > > $$
> > > we agree that explicitly plotting achieved velocity against the instructed target velocity provides a clearer view of instruction following.
> > >
> > > **How velocities are defined.**
> > > In the standard HalfCheetah-Vel setup, target forward velocities are generated from a fixed grid ranging from **0.075 m/s to 3.0 m/s**, using **uniform increments of 0.075 m/s**. A subset of these is used for training, while a separate set of **held-out evaluation velocities** is used for testing. The unseen test velocities we used—**0.225, 0.6, 1.2, 1.8, and 2.025 m/s**—come directly from this evaluation split.
> > >
> > > Following the reviewer’s suggestion, we instantiated TeNet policies from these unseen natural-language instructions and also evaluated an additional **out-of-range** instruction (3.5 m/s). We then compared the instructed versus achieved velocities.
> > > The achieved velocities are computed as the **average forward speed over the last 20 steps** of each rollout
> > >
> > > **Brief summary of the results.**
> > > Across all unseen evaluation velocities, TeNet-Contrast closely tracks the instructed speeds, confirming smooth generalization across the continuous velocity range. For the extrapolated instruction (3.5 m/s), both TeNet-Contrast and Prompt-DT converge to speeds near the upper end of the HalfCheetah dynamics (around ~3 m/s), reflecting the environment’s practical locomotion limit rather than a failure of instruction following.
> > >
> > > This analysis and the corresponding plot have been added to the **revised manuscript (Appendix C.6)**.

---

### Official Review · Reviewer_HthK · 2025-11-02

**Soundness:** 2
**Presentation:** 3
**Contribution:** 2
**Rating:** 4
**Confidence:** 2

**Summary:**

The paper introduces TENET, a framework for generating compact task-specific policy networks from language conditioned hypernetworks. In particular, the framework trains a hypernetwork conditioned on a natural language description of a task, which then outputs a set of parameters for a policy network. The language description is embedded with an LLM and further aligned with a trajectory encoding through a contrastive objective. The authors evaluate TENET on Mujoco gym environments and three Metaworld settings. Experiments show that TENET outperforms baselines while achieving high control frequency.

**Strengths:**

- Interesting Idea
- Good presentation
- Strong results: high-frequency, efficient policies.

**Weaknesses:**

The main weaknesses are misleading scope and overstated claims:
- The paper's core motivation, which frames TENET as complementary to large VLAs, is misleading, since VLAs operate in complex, high-diversity vision-based, real-world scenarios, whereas TENET is evaluated only in state-based, simulation environments.
- The claim that the framework makes “TeNet more scalable and practical for diverse task sets.” is not supported by experiments
- The language instruction and task diversity are very limited and seem to only differ in numerical target values such as target location or velocity. Language instructions might actually not be needed and could be replaced by a one hot vector or similar task encodings. Ablations in that regard are missing. Also, comparisons against task-embedding or archive-based methods are missing.
- In general, more complex experiment settings such as generalization to new objects or different tasks that actually require a semantic understanding of the task instruction would support the claim that the architecture applicable to  diverse tasks.
- Actual language instructions are not shown and task settings for MT are not described in enough detaill

TENET is a promising research direction, but its claims are currently overstated. The comparison to VLAs is not appropriate, and the work requires substantial new experiments in more challenging and diverse settings to prove that the language component provides a significant, generalizable advantage over simpler task identifiers. Without this evidence, the paper's novel contribution is limited.

**Questions:**

- How does this scale to different model sizes? Such a high control frequency is usually not needed (or usable) in downstream robot manipulation applications.
- Does the hypernetwork actually understand language instructions? How do different instructions change behavior?
- How does the approach compare against methods using task embeddings/one hot embeddings?
- How do different (smaller) language encoders influence performance?
- How does the architecture perform on IL-only benchmarks (without constrastive alignment of language and trajectory)?

---

> ### Author Response · Authors · 2025-11-18
> **Rebuttal to Reviewer HthK**
>
> We thank the reviewer for the constructive feedback and for noting the strengths of our work: the interesting idea, clear presentation, and strong empirical performance with high-frequency compact controllers. Below we address each concern point-by-point.
>
> ---
>
> ## 1. Scope and comparison to VLAs
>
> **Reviewer concern:**
> *The comparison to VLAs is misleading because VLAs operate in complex vision-based real-world settings, while TeNet is evaluated in state-based simulation.*
>
> **Response:**
> We agree that TeNet does not aim to solve vision-based robotic tasks, nor to replace VLAs. Our intent was only to highlight the **computational gap** between:
>
> - large VLA models (PaLM-E, RT-2, OpenVLA), and
> - lightweight controllers suitable for high-frequency onboard control.
>
> We will make this explicit in the introduction: TeNet is **complementary** to VLAs, not competitive with them. Evaluating in state-based simulation is an intentional first step toward investigating compact, language-conditioned policy synthesis.
>
> ---
>
> ## 2. Task diversity and the role of language
>
> **Reviewer concern:**
> *The language instructions and task diversity are very limited and seem to differ only in numerical values. Language might not be needed and could be replaced by one-hot identifiers.*
>
> **Response:**
> A key motivation of TeNet is to enable **natural-language-driven policies** that are both compact and fast. Natural language is not used merely to restate canonical task templates (e.g., “Reach the target position.”), but to allow the agent to understand **different levels of paraphrasing**, such as:
>
> - “Bring the end effector all the way to the target location without interacting with any objects.”
> - “Your goal is to drive the end effector toward the marked target and stop exactly when you arrive.”
>
> TeNet handles such paraphrasing effectively. As shown in **Appendix C.4**, using 1, 2, 5, or 10 paraphrases per task gives nearly identical performance, demonstrating that the language encoder captures meaningful structure rather than acting as a numeric label.
>
> Regarding task diversity, our benchmarks contain **substantial semantic variation**:
>
> - **MT10/MT50** contain **10 and 50 distinct manipulation skills** (drawer-open, hammer-pull, peg-insert, button-press, shelf-place, etc.), each requiring different behaviors, object affordances, and motor strategies.
> - **Ant-Dir** and **HalfCheetah-Vel/Dir** contain **continuous task distributions**, which cannot be represented or interpolated by one-hot IDs.
> - **ML1 Pick-Place** requires generalization to **unseen 3D object–goal configurations**.
>
> Natural language further enables **zero-shot description of entirely unseen tasks**, which one-hot/task-ID conditioning cannot support.
>
> ---
> ## 3. Missing ablations (one-hot, learned embeddings, smaller encoders)
>
> **Reviewer concern:**
> *Ablations comparing against simple task encodings (one-hot, learned embeddings, smaller encoders) are missing.*
>
> **Response:**
> Our experiments already include several conceptual baselines directly relevant to this concern:
>
> - **Direct TeNet**: evaluates the use of raw text embeddings without grounding
> - **TeNet-MSE / TeNet-Contrast**: ablate different grounding strategies
> - **Prompt-DT-HN**: serves as a strong non-language hypernetwork baseline
>
> **Prompt-DT as task identifier.**
> Prompt-DT uses a short demonstration at test time as a **task identifier**, encoding the task’s goal and behavioral pattern. As task diversity increases, these identifiers become harder to disambiguate. This is reflected in our results: Prompt-DT’s performance drops from **0.73 (MT10)** to **0.61 (MT50)** (Table 1), despite identical architectures and training conditions.
> TeNet avoids this limitation by conditioning directly on natural language and requiring no demonstrations at test time.
>
> **Archive-based approaches.**
> Archive-based policy methods (e.g., skill libraries) are conceptually orthogonal to our setting: TeNet uses a *single* hypernetwork that directly generates policy parameters from language, rather than maintaining large archives of task-specific controllers. A detailed comparison to archive-based methods is interesting future work but beyond the scope of this paper.
>
> **Meta-learning limitation.**
> Task-ID approaches—whether one-hot or prompt-based—cannot generalize to **unseen tasks** without introducing new identifiers. In contrast, TeNet can instantiate policies directly from natural-language descriptions, enabling zero-shot generalization in ML1, Ant-Dir, and HalfCheetah-Vel.
>
> We will make these distinctions clearer in the revised manuscript.

---

> ### Author Response · Authors · 2025-11-18
> **Rebuttal to Reviewer HthK (continued)**
>
> ## 4. Examples of language instructions and MT task settings
>
> **Reviewer concern:**
> *Task instructions are not shown, and MT settings are insufficiently detailed.*
>
> **Response:**
> We will add explicit examples to Appendix A. Representative instructions include:
>
> - **HalfCheetah-Vel:** “Move forward with target velocity 2.0 m/s.”
> - **Ant-Dir:** “Walk in the direction of 125 degrees.”
> - **ML1 Pick-Place:** “Pick up the block and place it at position (−0.1, 0.2, 0.1).”
> - **MT tasks:**
>   - “Open the sliding door.”
>   - “Pull the drawer open.”
>   - “Press the button on the wall.”
>   - “Push the block to the right side.”
>
> Appendix A will be expanded to clearly describe all task settings and their corresponding instructions.
>
> ---
>
> ## 5. Claim about scalability
>
> **Reviewer concern:**
> *The claim that TeNet is ‘more scalable and practical for diverse task sets’ is not sufficiently supported.*
>
> **Response:**
> We will soften this phrasing. Our intent was to highlight scalability in terms of:
>
> - **number of tasks** (up to 2,500 tasks in MT50),
> - **runtime efficiency** (40K-parameter controllers vs 1–39M in baselines),
> - **high control frequency** (>9 kHz).
>
> We do not claim scalability to vision-based or real-world domains in the present work.
>
> ---
>
> ## 6. Responses to reviewer’s clarification questions
>
> ### **(i) Scaling to different hypernetwork sizes**
> Hypernetwork size can be varied independently of the controller. Preliminary tests showed no consistent improvement with larger hypernetworks; we agree further exploration is valuable.
>
> ### **(ii) Does TeNet understand language?**
> Yes. Grounding experiments (Fig. 2, Sec. 5.4) show consistent behavioral differences across varying instructions and improved generalization through contrastive grounding.
>
> ### **(iii) Comparison to one-hot/task embeddings**
> Addressed in Sections 2–3. One-hot IDs cannot generalize to unseen tasks and become ambiguous in large multi-task settings.
>
> ### **(iv) Smaller language encoders**
> The choice of encoder does not affect control performance, as it is used **only once** outside the control loop. TeNet can operate with smaller encoders (e.g., BERT) without impacting control frequency.
>
> We use LLaMA mainly for **robustness to flexible natural-language instructions**. As shown in **Appendix C.4**, TeNet remains stable when using 1, 2, 5, or 10 paraphrases per task.
>
> We further conducted a direct encoder comparison on **MT10**, training with 10 simple (Level 0) paraphrases and testing on more challenging paraphrases:
>
> - **Level 1:** multi-clause, mild context (~12–30 tokens)
> - **Level 2:** narrative-style, meta-phrases, extra information (~20–60 tokens)
>
> **Results on MT10:**
>
> | Encoder | Level 0 | Level 1 | Level 2 |
> |---------|---------|---------|---------|
> | **LLaMA** | 0.99 | 0.95 | 0.89 |
> | **BERT** | 0.99 | 0.89 | 0.82 |
>
> Both encoders perform similarly under simple paraphrasing, but LLaMA is substantially more robust under **medium** and **noisy** paraphrastic variations.
> These results have been added to **Appendix C.5**.
>
> ### **(v) IL-only variant**
> Direct TeNet corresponds to this setting; Fig. 2 shows that grounding improves performance but is not required for viability.
>
> ---
>
> # Summary
>
> We will revise the paper to:
>
> - clarify TeNet’s scope relative to VLAs,
> - expand task descriptions and MT settings,
> - temper the scalability claim,
> - discuss simple task encodings more explicitly,
> - and highlight the benefits of flexible, robust natural-language task specification.
>
> We thank the reviewer again for the thoughtful feedback.

---

### Author Response · Authors · 2025-11-18
**Summary of Manuscript Revisions**

All manuscript updates are highlighted in **blue** in the revised manuscript.
This document summarizes all revisions made in response to reviewer feedback, grouped by theme.
Each item lists: (i) what changed, (ii) where it appears, and (iii) which reviewers it addresses.

---

## 1. Clarifying Scope and Relation to VLAs
**Addresses:** HthK, sMsz, 8zws

**What changed:**
- Clarified that TeNet operates **exclusively in state-based simulation**.
- Stated clearly that TeNet is **complementary to VLAs**, not competitive.
- Softened all scalability/practicality claims.

**Where:**
Abstract; Introduction (paragraphs 2–3 and final paragraph); Section 5.2; Discussion.

---

## 2. Clarifying the Main Contribution
**Addresses:** 8zws, HthK, sMsz

**What changed:**
- Emphasized that natural language is used **only once** to instantiate a ~40K-parameter controller.
- Highlighted that this controller runs **without any LLM/VLM** at inference.

**Where:**
Introduction → Contribution bullets (first bullet).

---

## 3. Grounding Is Auxiliary
**Addresses:** 8zws, HthK, sMsz

**What changed:**
- Stated explicitly that MSE/contrastive alignment is a **standard tool**, not a novelty.
- Clarified that TeNet’s core novelty is **text-conditioned hypernetwork policy generation**.
- Added explanation relating deterministic alignment to CLASP’s distributional encoders.

**Where:**
Introduction → Contribution bullets (second bullet);
Related Work (CLASP paragraph);
Section 4.3.

---

## 4. Expanded Related Work: Make-an-Agent & Diffusion Policies
**Addresses:** H4RM, sMsz

**What changed:**
- Added discussion of **Make-an-Agent** as a trajectory-conditioned policy generator.
- Added a new paragraph discussing **Diffusion Policy**, **MTDiff (vision)**, and **MetaDiffuser (vision)**.
- Clarified why diffusion-based visuomotor policies are **not direct baselines** (vision-based, different encoder).

**Where:**
Related Work → Hypernetworks;
Related Work → Compact Sequence Models (final blue paragraph).

---

## 5. Clarifying Why Simple Task Identifiers Are Insufficient
**Addresses:** HthK, 8zws

**What changed:**
- Added explanation of why one-hot IDs / task embeddings cannot support natural-language generalization or unseen tasks.

**Where:**
Related Work → Hypernetworks (task-embedding discussion).

---

## 6. Encoder Size Rationale (LLaMA vs. BERT)
**Addresses:** sMsz, HthK

**What changed:**
- Added explanation that the text encoder is called **once per task**, so size has **zero runtime impact**.
- Added statement that **smaller encoders** also work.
- Referenced paraphrasing robustness results (Appendix C.5).

**Where:**
Appendix B.1 (Text Encoder — final paragraph);
Section 5.8.

---

## 7. Task Instructions & Paraphrasing Diversity
**Addresses:** HthK

**What changed:**
- Added explicit examples of natural-language instructions for all tasks.
- Added description of paraphrasing levels (Level 0/1/2).
- Clarified paraphrasing robustness evaluation.

**Where:**
Appendix A.4;
Appendix C.4;
Appendix C.5.

---

## 8. Prompt-DT-HN Identified as a Trajectory-Conditioned Hypernetwork Baseline
**Addresses:** H4RM

**What changed:**
- Clarified that Prompt-DT-HN corresponds directly to a **trajectory-based hypernetwork** baseline.

**Where:**
Section 5.5;
Appendix B.3.

---

## 9. Trajectory Encoder Training & Real-World Deployment
**Addresses:** H4RM

**What changed:**
- Clarified that the trajectory encoder is **trained jointly**, not pretrained separately.
- Added short note on real-world deployment: demonstration noise, domain shift, perception requirements.

**Where:**
Appendix B.1;
Discussion.

---

## 10. Softened Scalability Claims
**Addresses:** HthK, 8zws, sMsz

**What changed:**
- Replaced all broad scalability claims with precise references to **state-based, offline imitation** settings.

**Where:**
Section 5.2;
Discussion;
Introduction (final paragraph).

---

## 11. Added Velocity–Following Analysis (Appendix C.6)
**Addresses:** H4RM, 8zws

**What changed:**
- Added a dedicated analysis of TeNet’s instruction–velocity alignment in HalfCheetah-Vel.
- Clarified how target velocities are defined in the benchmark (0.075–3.0 m/s in 0.075 m/s steps).
- Provided a brief explanation of results showing that TeNet tracks unseen velocity instructions smoothly and saturates near the environment’s practical speed limit.

**Where:**
Appendix C.6.

---

# Final Result

The revised manuscript now:

- **Directly addresses all reviewer concerns**,
- Avoids any overstated novelty claims,
- Clarifies the intended scope (state-based, not vision),
- Presents TeNet as a **complementary paradigm** rather than a VLA competitor,
- Strengthens related work and architectural justifications,
- And improves transparency around encoder choice, grounding, and baselines.

---

### Author Response · Authors · 2025-11-28
**Request for Feedback on Addressed Concerns**

We thank all reviewers for the constructive discussion so far. As the discussion period is nearing its end, we would like to kindly ask whether our responses have fully addressed your concerns, or if any ambiguities or additional clarifications remain. We are happy to provide further details or revisions if needed, and we greatly appreciate your responses as they help us further improve the paper.

---

### Note · Authors · 2026-01-20

**Comment:**

The authors have decided to withdraw this submission due to unforeseen circumstances that prevented completion of the rebuttal process. We thank the reviewers and area chair for their time and feedback.

**Withdrawal Confirmation:**

I have read and agree with the venue's withdrawal policy on behalf of myself and my co-authors.